# Dynamic NF-κB and E2F interactions control the priority and timing of inflammatory signalling and cell proliferation

John M Ankers[1], Raheela Awais[1,2†], Nicholas A Jones[2†], James Boyd[2], Sheila Ryan[1,2], Antony D Adamson[2], Claire V Harper[2], Lloyd Bridge[2,3], David G Spiller[2], Dean A Jackson[2], Pawel Paszek[2], Violaine Sée[1], Michael RH White[2*]

[1]Centre for Cell Imaging, Institute of Integrative Biology, Liverpool, United Kingdom; [2]Systems Microscopy Centre, Faculty of Life Sciences, Manchester, United Kingdom; [3]Department of Mathematics, University of Swansea, Swansea, United Kingdom

*For correspondence: mike.white@manchester.ac.uk

†These authors contributed equally to this work

Competing interests: The authors declare that no competing interests exist.

**Abstract** Dynamic cellular systems reprogram gene expression to ensure appropriate cellular fate responses to specific extracellular cues. Here we demonstrate that the dynamics of Nuclear Factor kappa B (NF-κB) signalling and the cell cycle are prioritised differently depending on the timing of an inflammatory signal. Using iterative experimental and computational analyses, we show physical and functional interactions between NF-κB and the E2 Factor 1 (E2F-1) and E2 Factor 4 (E2F-4) cell cycle regulators. These interactions modulate the NF-κB response. In S-phase, the NF-κB response was delayed or repressed, while cell cycle progression was unimpeded. By contrast, activation of NF-κB at the G1/S boundary resulted in a longer cell cycle and more synchronous initial NF-κB responses between cells. These data identify new mechanisms by which the cellular response to stress is differentially controlled at different stages of the cell cycle.

## Introduction

One of the most important functions in a cell is the accurate interpretation of the information encoded in extracellular signals leading to context-dependent control of cell fate. This is achieved via complex and dynamic signal transduction networks, through which gene expression is re-programmed in response to specific environmental cues (*Barabási et al., 2011*). Many signalling systems are subject to temporal changes, involving dynamic alterations to the states of their constituent genes and proteins, with time scales ranging from seconds (Calcium signalling [*Berridge, 1990*; *Schmidt et al., 2001*]), to hours (DNA damage response [*Lahav et al., 2004*], inflammatory response [*Ashall et al., 2009*]), to days (circadian clock [*Welsh et al., 2004*], cell cycle [*Sakaue-Sawano et al., 2008*]). Although previous studies have indicated interactions between proteins associated with different dynamical systems (*Wilkins and Kummerfeld, 2008*; *Bieler et al., 2014*; *Feillet et al., 2014*), how and when signalling systems are dynamically integrated to determine important cell fate decisions is not well understood.

Nuclear Factor kappa B (NF-κB) is an important signalling system, implicated in many diseases including autoimmune diseases and cancer (*Grivennikov et al., 2010*). Inflammatory cues such as Tumour Necrosis Factor alpha (TNFα) can trigger the nuclear translocation of the NF-κB RelA subunit and activation of target gene transcription (*Hayden and Ghosh, 2008*). Nuclear NF-κB activates

**eLife digest** Investigating how cells adapt to the constantly changing environment inside the body is vitally important for understanding how the body responds to an injury or infection. One of the ways in which human cells adapt is by dividing to produce new cells. This takes place in a repeating pattern of events, known as the cell cycle, through which a cell copies its DNA (in a stage known as S-phase) and then divides to make two daughter cells. Each stage of the cell cycle is tightly controlled; for example, a family of proteins called E2 factors control the entry of the cell into S phase.

"Inflammatory" signals produced by a wound or during an infection can activate a protein called Nuclear Factor-kappaB (NF-κB), which controls the activity of genes that allow cells to adapt to the situation. Research shows that the activity of NF-κB is also regulated by the cell cycle, but it has not been clear how this works. Here, Ankers et al. investigated whether the stage of the cell cycle might affect how NF-κB responds to inflammatory signals.

The experiments show that the NF-κB response was stronger in cells that were just about to enter S-phase than in cells that were already copying their DNA. An E2 factor called E2F-1 –which accumulates in the run up to S-phase – interacts with NF-κB and can alter the activity of certain genes. However, during S-phase, another E2 factor family member called E2F-4 binds to NF-κB and represses its activation. Next, Ankers et al. used a mathematical model to understand how these protein interactions can affect the response of cells to inflammatory signals.

These findings suggest that direct interactions between E2 factor proteins and NF-κB enable cells to decide whether to divide or react in different ways to inflammatory signals. The research tools developed in this study, combined with other new experimental techniques, will allow researchers to accurately predict how cells will respond to inflammatory signals at different points in the cell cycle.

feedback regulators, including the inhibitory kappa B alpha (IκBα) and epsilon (IκBε) inhibitors (*Arenzana-Seisdedos et al., 1997*; *Kearns et al., 2006*; *Paszek et al., 2010*), which bind and transport NF-κB back into the cytoplasm. In response to TNFα, this system shows nuclear-cytoplasmic (N:C) oscillations in the localization of the NF-κB complex associated with out-of-phase cycles of degradation and re-synthesis of IκB proteins (*Nelson et al., 2004*; *Ashall et al., 2009*; *Lee et al., 2009*; *Sung et al., 2009*; *Tay et al., 2010*; *Turner et al., 2010*; *Ruland, 2011*; *Hughey et al., 2015*). Through systems biology and experimental approaches, the frequency of these oscillations has been proposed to be a key parameter that regulates the pattern of downstream gene expression (*Ashall et al., 2009*; *Lee et al., 2014*; *Williams et al., 2014*).

NF-κB signalling has also been suggested to have a role in controlling cell division through a number of different mechanisms (*Perkins and Gilmore, 2006*). Many NF-κB family members have been characterised as oncoproteins (e.g. c-Rel and Bcl-3 [*Hayden and Ghosh, 2008*]). Also, a number of cell cycle control proteins have been shown to be NF-κB transcriptional targets, including Cyclin D, (*Guttridge et al., 1999*; *Sée et al., 2004*) and p21, an inhibitor of Cyclin Dependent Kinase (CDK) activity (*Hinata et al., 2003*).

Although interactions between NF-κB and the cell cycle have been reported (*Kundu et al., 1997*; *Phillips et al., 1999*; *Perkins and Gilmore, 2006*); observing the dynamics of such interactions is challenging via traditional biochemical techniques, which often fail to capture the heterogeneity in a cellular population. Analysis of cell-to-cell heterogeneity has revealed novel regulatory mechanisms for diverse cellular processes (*Pelkmans, 2012*) and it has been suggested that this is a fundamental property of the NF-κB response (*Paszek et al., 2010*).

The E2 Factor (E2F) proteins are differentially expressed during the cell cycle to control cell proliferation (*Bertoli et al., 2013*). They are a family of transcription factors that play a key role in the G1/S cell cycle checkpoint. Previous studies have provided preliminary evidence for physical interaction between NF-κB and E2F proteins (*Tanaka et al., 2002*; *Lim et al., 2007*; *Garber et al., 2012*) In the current study, a combination of single cell imaging and mathematical modelling was applied to

investigate reciprocal co-ordination of the NF-κB response and cell proliferation driven by dynamic interactions between RelA and E2F proteins.

## Results

### The NF-κB response depends on the cell cycle phase

We investigated the effect of cell cycle timing on the NF-κB response in HeLa cervical cancer and SK-N-AS neuroblastoma cells. SK-N-AS cells showed repeated oscillations in response to TNFα stimulation that were more damped than those seen in HeLa cells (see *Appendix 1—figure 1* for longer time course data [*Nelson et al., 2004*; *Ashall et al., 2009*]). In previous studies it was observed that when SK-N-AS cells were treated with a saturating dose of TNFα (10 ng/ml) the initial response of NF-κB (i.e. immediate RelA nuclear translocation) was relatively synchronous between cells (*Nelson et al., 2004*; *Ashall et al., 2009*; *Turner et al., 2010*) (*Figure 1A*; *Appendix 1—figure 1*). However, these data showed a variation in timing and amplitude when cells were treated with a lower dose of 30 pg/ml TNFα, even though this was functionally close to a saturating dose that gave a strong population-level NF-κB response (*Turner et al., 2010*) (*Figure 1B*). In common with treatment of SK-N-AS cells at 30 pg/ml, HeLa cells showed greater heterogeneity in their initial response at a saturating 10 ng/ml dose of TNFα, with some cells showing little or no response and others showing a variable delay (*Nelson et al., 2002*; *2004*) (*Figure 1C*). This is in agreement with data showing heterogeneity of the initial response in other cell types (*Tay et al., 2010*; *Zambrano et al., 2014*). HeLa cells showed no significant translocation in response to 30 pg/ml TNFα (*Figure 1D*), suggesting that these cell types have differential dynamic NF-κB responses at varying TNFα doses.

We hypothesised that this cell-to-cell heterogeneity in response might be a consequence of cell cycle phase. To test this hypothesis, we investigated the role of cell cycle in both the HeLa and SK-N-AS cells, as these show different dynamic responses to TNFα that are typical of the profile of a wide range of cell lines (*Tay et al., 2010*; *Turner et al., 2010*; *Zambrano et al., 2014*; *Hughey et al., 2015*). Initially, HeLa cells were treated with 10 ng/ml TNFα at various stages of the cell cycle (*Figure 1E–H*), as they could be easily synchronized at late G1 by a double thymidine block (see *Appendix 1—figure 2*). When endogenous RelA was examined using immunocytochemistry, HeLa cells treated with 10 ng/ml TNFα in S-phase displayed a reduced nuclear localization, compared to those treated in late G1 (*Figure 1E*). These results were confirmed using time-lapse imaging of synchronised HeLa cells transiently transfected with RelA-DsRedxp. Cells treated in late G1 showed a strong synchronous translocation of RelA, whereas cells treated in S-phase showed reduced RelA translocation (*Figure 1F,G*). These cell cycle-dependent differences following TNFα treatment of synchronized cell populations were supported by alterations in the extent of IκBα degradation and RelA Serine$^{536}$ phosphorylation at different stages of the cell cycle as measured by western blot (*Figure 1H*).

To further investigate the effect of cell cycle on the NF-κB response, unsynchronized populations of HeLa and SK-N-AS cells were followed by time-lapse imaging through successive cell divisions. 30 hr after the start of this time-course, HeLa cells were stimulated with 10 ng/ml TNFα. Cells were assigned to different cell cycle phases based upon their mitosis-to-mitosis and mitosis-to-treatment timings (*Figure 2A and B*).

To ensure the accuracy of the inferred cell cycle stage in these experiments, the cycle timing of cells at the point of TNFα treatment was calibrated through control experiments using Fluorescent Ubiquitin-based Cell Cycle Indicators (FUCCI) in both HeLa and SK-N-AS cells (*Figure 2—figure supplement 1A–B*). The crossing point of Red and Green FUCCI reporters was determined, and defined as the G1/S checkpoint. The average and distribution of the cell cycle duration in populations of HeLa and SK-N-AS cells was also measured (*Figure 2—figure supplement 1C*).

The resulting data suggested that HeLa cells treated with TNFα in late G1 (inferred to be G1/S) showed an increase in the translocation amplitude compared to the unsynchronized population average (*Figure 2C*). By contrast, cells treated in S-phase appeared to show a damped or delayed response (*Figure 2C*), with markedly reduced amplitude of nuclear NF-κB translocation. In G2 phase the NF-κB response appeared to be restored. Analysis of the complete data set confirmed that there was statistically significant higher nuclear translocation amplitude in HeLa cells at G1/S and significantly reduced amplitude in S-phase, compared to G1 and G2 (*Figure 2—figure supplement 2*).

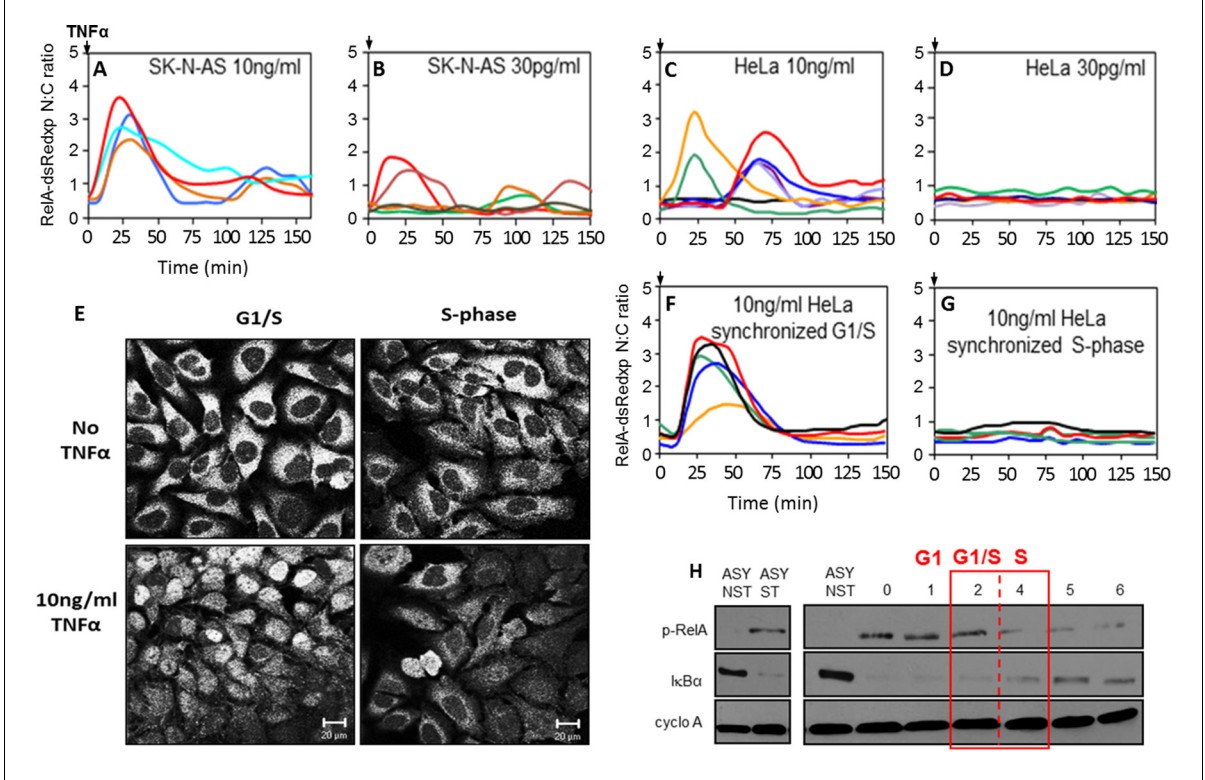

**Figure 1.** NF-κB dynamics following TNFα treatment in HeLa and SK-N-AS cells: Mapping the NF-κB response over the cell cycle in synchronized HeLa cells. (**A,B,C** and **D**) The dynamics of RelA-dsRedxp following 10 ng/ml TNFα treatment in transiently transfected SK-N-AS (**A**), or following 30 pg/ml TNFα treatment in SK-N-AS, and 10 ng/ml TNFα treatment in HeLa cells (**C**), and at 30 pg/ml for HeLa (**D**) cells (n=30 cells analysed per condition). (**E**) The localization of endogenous RelA in different cell cycle phases, observed by immunocytochemistry at 2 hr (G1/S transition), 4 hr (mid S-phase), post-release from double thymidine block and with 15 min TNFα treatment. (**F** and **G**) The dynamics of RelA-dsRedxp in transiently transfected HeLa cells synchronized by a double thymidine block, following 10 ng/ml TNFα treatment at G1/S (**F**), or passing through S-phase (**G**) (n=20 cells analysed per condition). (**H**) Western blot of Ser$^{536}$phopho-RelA (p-RelA), IκBα, and cyclophilin-A (cyclo-A) levels in synchronized HeLa cells harvested at 1 hr time intervals over the G1/S transition following 15 min treatment with TNFα. Also shown are asynchronous, non-stimulated (ASY NST) and asynchronous, stimulated (ASY ST) controls, harvested at t=0.

A smaller data set from SK-N-AS cells treated with 30 pg/ml TNFα, showed once again a statistically reduced translocation in S-phase compared to G1-phase. Visually the data are consistent with increased translocation in late G1 and a restored level of translocation in G2- compared to S-phase. However more cells would be required for a statistical analysis of possible differences between these cell cycle phases. (*Figure 2—figure supplement 3*).

## The effect of NF-κB signalling on cell cycle timing

We also measured the effect of TNFα treatment on HeLa cell cycle duration (*Figure 3*). It was found that mean cell cycle duration for cells treated with TNFα showed a small, but statistically significant increase of 1.9 hr (~10%) compared to untreated cells, with the variability in the total population increasing by ~2-fold (*Figure 3*). Within this TNFα-treated population, cells treated in late G1 were more susceptible to cell cycle elongation with a cell cycle duration that was ~1/3 longer than the untreated population average. TNFα treatment in S-phase had no statistically significant effect on the timing of mitosis. These data suggest a potential direct or indirect role for the NF-κB system in controlling cell cycle duration through an unknown mechanism at the G1/S phase of the cell cycle.

## E2F-1 levels control the dynamics of the NF-κB response

The mechanism for alteration of NF-κB responses between the late G1- and S-phases of the cell cycle was sought. Previous studies had suggested that E2-Factor-1 (E2F-1) could physically associate

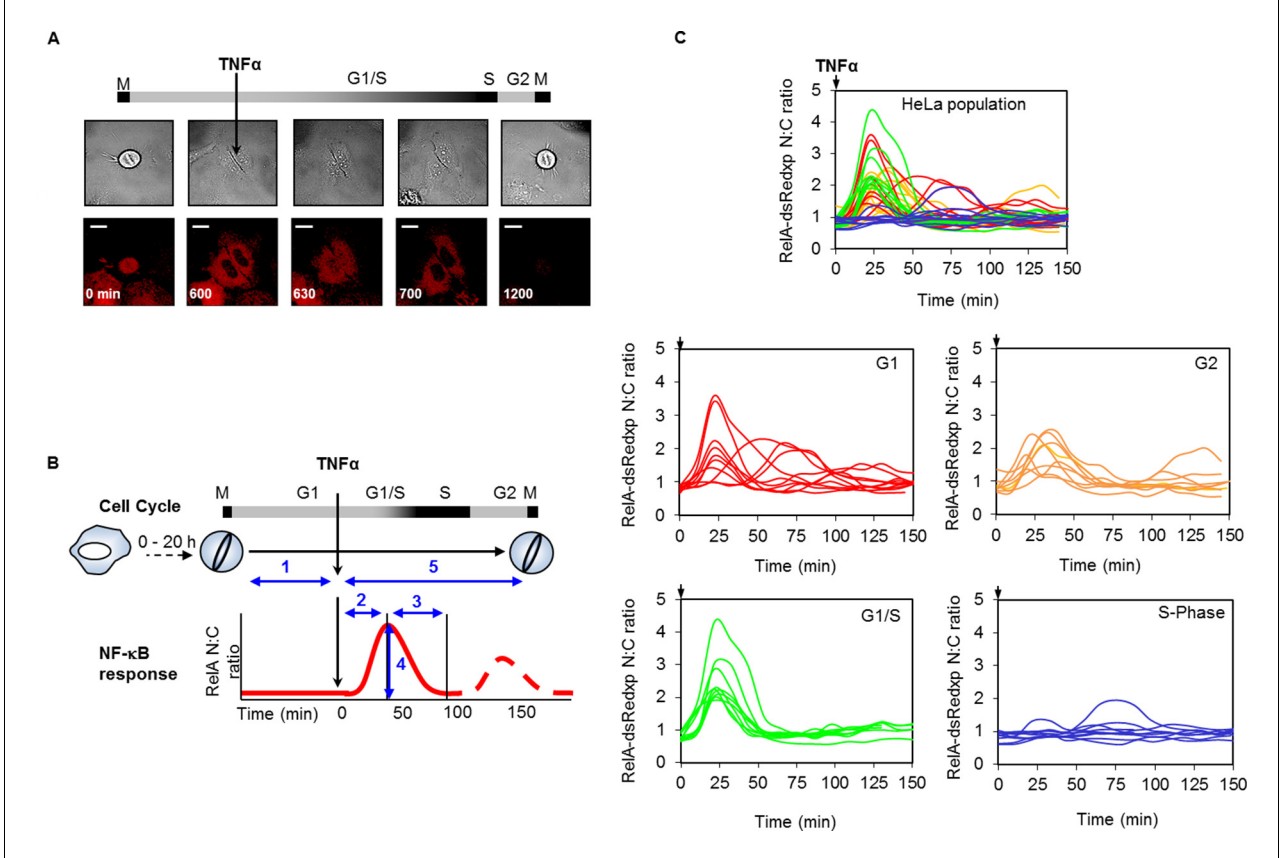

**Figure 2.** Mapping the NF-κB response over the cell cycle through virtual synchronization. (**A**) Selected images from time-lapse imaging of RelA-dsRedxp transiently expressing Hela cells treated with 10 ng/ml TNFα. (**B**) Virtual synchronization of HeLa cells treated with 10 ng/ml TNFα. Cells were imaged through two successive divisions (M) allowing correlation of cell cycle timing of TNFα treatment (parameter 1) to RelA dynamics (parameters 2, 3 and 4) and cell cycle duration (parameters 1 plus 5). (**C**) Representative cells of RelA-dsRedxp dynamics following TNFα treatment in asynchronous cells, then virtually synchronized into G1 (n=115), G1/S (n=32), S (n=52) and G2 (n=38) phases.

The following figure supplements are available for figure 2:

**Figure supplement 1.** Analysis of cell cycle duration and G1/S timing in HeLa and SK-N-AS cells.

**Figure supplement 2.** Statistical analysis of NF-κB translocation in HeLa cells at inferred cell cycle stages following 10 ng/ml TNFα stimulation.

**Figure supplement 3.** Statistical analysis of NF-κB translocation in SK-N-AS cells at inferred cell cycle stages following 30 pg/ml TNFα stimulation.

with RelA, and/or its major dimer partner p50 (*Kundu et al., 1997*; *Tanaka et al., 2002*; *Lim et al., 2007*). E2F-1 is the key transcriptional regulator of the cell cycle transition between G1- and S-phase (*Tsantoulis and Gorgoulis, 2005*), where its expression is highest. In the presence of ectopically-expressed EGFP-E2F-1, we observed a reduction in the activity of a NF-κB-regulated luciferase reporter (*Figure 4A*). Moreover, the ability of NF-κB to induce endogenous mRNA levels of IκBα and IκBε was impaired in cells co-expressing EGFP-E2F-1 and RelA-DsRedxp, compared to cells expressing RelA-DsRedxp alone (*Figure 4B*). E2F-1 target gene transcription was also impaired by RelA expression, as indicated by a reduction in the activity of a Cyclin E luciferase reporter (*Figure 4C*) and in the mRNA level of E2F-1 itself (*Figure 4D*). These data support the reciprocal and coordinated control of transcription by E2F-1 and NF-κB.

In transient transfection experiments, a predominantly cytoplasmic localization of RelA-DsRedxp was observed when expressed alone, whereas in cells co-expressing EGFP-E2F-1, both proteins were predominantly nuclear (*Figure 4E*). In addition we also found that the steady-state cytoplasmic localisation of RelA was restored in cells transiently expressing IκBα-AmCyan in addition to EGFP-

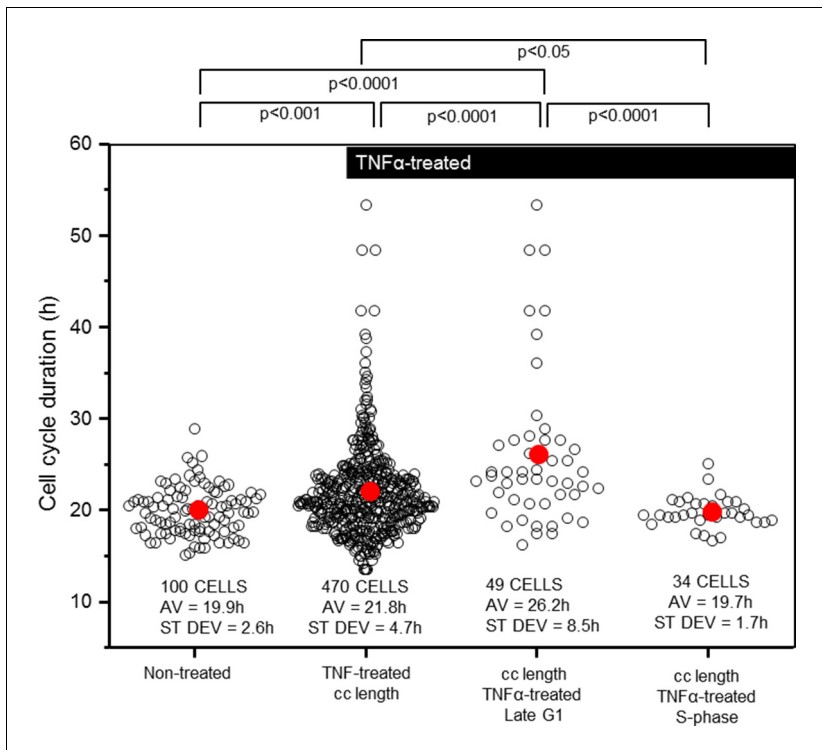

**Figure 3.** Cell cycle length and variability is modified by TNFα addition at G1/S. Analysis of the timing and variability of mitosis (parameter 1 plus 5 from **Figure 2B**) following 10 ng/ml TNFα treatment of asynchronous untransfected HeLa cells, compared to subsets of those cells stimulated at late G1- or S-phase. Mean durations were analysed using nonparametric Anova analysis with Dunn correction for multiple comparisons. Variability in the data was analysed using Levene's test for equality of variance.

E2F-1 and RelA-dsRedxp. These data suggest the hypothesis that IκBα and E2F-1 may compete for the same binding site on RelA, with IκBα perhaps having the higher affinity. Time-series experiments in both SK-N-AS and HeLa cells showed that a decrease in EGFP-E2F-1 expression over time was associated with a re-localization of RelA-DsRedxp from the nucleus to the cytoplasm (for SK-N-AS cells, *Figure 4—figure supplement 1A–B*; for HeLa cells, *Figure 4—figure supplement 2A–C*). Quantitative analysis showed a strong correlation between the EGFP-E2F-1 decay half-life and the delay in RelA-DsRedxp translocation back into the cytoplasm (for SK-N-AS cells, *Figure 4—figure supplement 1C*; for HeLa cells, *Figure 4—figure supplement 2D*). Initial mathematical modelling of this interacting system (for details of the model see Appendix Section B) was able to recapitulate the main features of the observed correlation between E2F-1 levels and RelA localization in silico (*Figure 4—figure supplement 1D–E*).

## Physical and functional interaction between RelA and E2F-1

These data supported a direct interaction between E2F-1 and RelA. Therefore, the physical interactions between E2F-1 and NF-κB proteins in cells were investigated. Co-localization of E2F-1 and RelA had previously been shown through fluorescence imaging experiments (see *Figure 5A*). A clear physical interaction between fluorescently labelled E2F-1 and RelA in the nucleus of living cells was evident using Förster Resonance Energy Transfer (FRET), in conjunction with acceptor photobleaching as a qualitative indicator of intermolecular interaction (*Figure 5D*), and Fluorescence Cross-Correlation Spectroscopy (FCCS) (*Figure 5C*).

In order to further support the interaction between the endogenous proteins, we used co-immunoprecipitation (Co-IP) of endogenous E2F-1 and RelA in HeLa cells that had been synchronized in late G1, when E2F-1 levels were at their peak (*Figure 5B*). These data confirmed a physical interaction between E2F-1 and RelA, in agreement with previous studies (*Tanaka et al., 2002*; *Lim et al.,*

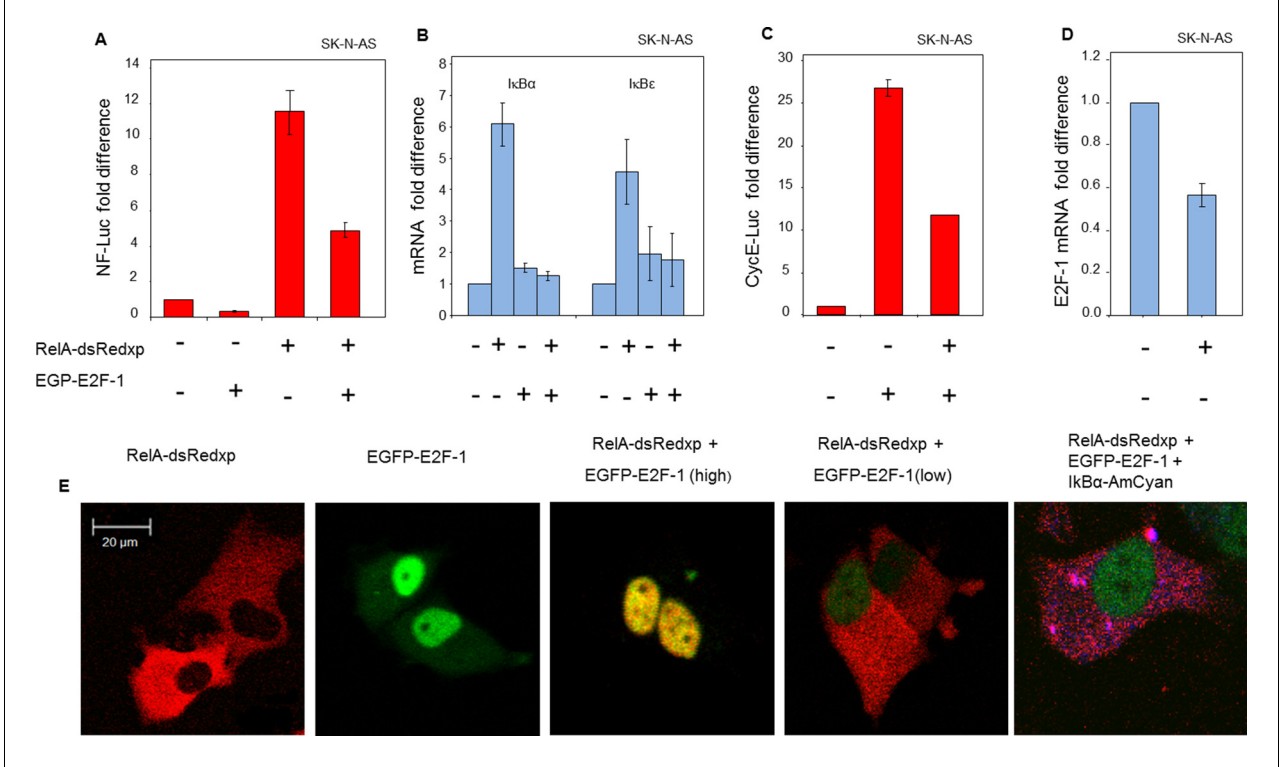

**Figure 4.** Physical and functional interaction between NF-κB and E2F-1 systems. (A) NF-κB-dependent transcription was assessed by luciferase reporter assay (NF-luc), in SK-N-AS cells (n=3, +/- s.d) expressing EGFP-E2F-1, RelA-dsRedxp or both. (B) IκBα and IκBε mRNA levels in SK-N-AS cells (n=3, +/- s.d) following transient expression of EGFP-E2F-1, RelA-DsRedxp or both. (C) E2F-1-dependent transcription as assessed by luciferase reporter assay (CyclinE-luc), in SK-N-AS cells (n=3, +/- s.d) expressing EGFP-E2F-1, RelA-dsRedxp or both. (D) E2F-1 mRNA levels in SK-N-AS cells (n=3, +/- s.d) transiently transfected with RelA-dsRedxp. (E) Representative SK-N-AS cells transiently expressing EGFP-E2F-1 (green), RelA-dsRedxp (red), both fluorescent fusion proteins at different levels, or EGFP-E2F-1, RelA-dsRedxp and IκBα-AmCyan (blue).

The following figure supplements are available for figure 4:

**Figure supplement 1.** E2F-1 modulates NF-κB dynamics in the absence of stimulus in SK-N-AS cells.

**Figure supplement 2.** E2F-1 modulates NF-κB dynamics in the absence of stimulus in HeLa cells.

*2007; Garber et al., 2012*). We were not able to observe a positive co-IP in asynchronous cells (see *Appendix 1—figure 4*), suggesting that this interaction was only detectable in HeLa cells at G1/S when E2F-1 was at its highest level. Considered together, all of these different measurements support a significant interaction between these proteins. These data suggest the hypothesis that the interaction between RelA and E2F-1 in the nucleus of G1/S cells, which have been subjected to an inflammatory stimulus, may coordinate differential regulation of NF-κB target gene transcription.

## In-silico modelling and prediction of NF-κB interaction with E2F-4

In order to understand and further investigate the dynamic behaviour of TNF-α-mediated NF-κB activation in the presence of E2F-1 (at the G1-S transition), an ordinary differential equation-based mathematical model of the NF-κB system (*Ashall et al., 2009*) was extended to include the interaction with E2F-1 (see Appendix Section B). In this model, E2F-1 was assumed to compete with IκBα for binding to free NF-κB, but had no effect on the localization of RelA bound to IκBα. Simulations (of nuclear NF-κB levels over time from transfection experiments) using this model, supported the hypothesis that E2F-1 might temporally control the duration of RelA nuclear occupancy through a combination of binding to RelA in the nucleus and inhibition of RelA-dependent IκBα transcription (as suggested by data shown in *Figure 4*). E2F-1 degradation could allow NF-κB to re-activate IκBα, which in turn could restore RelA to a cytoplasmic localization.

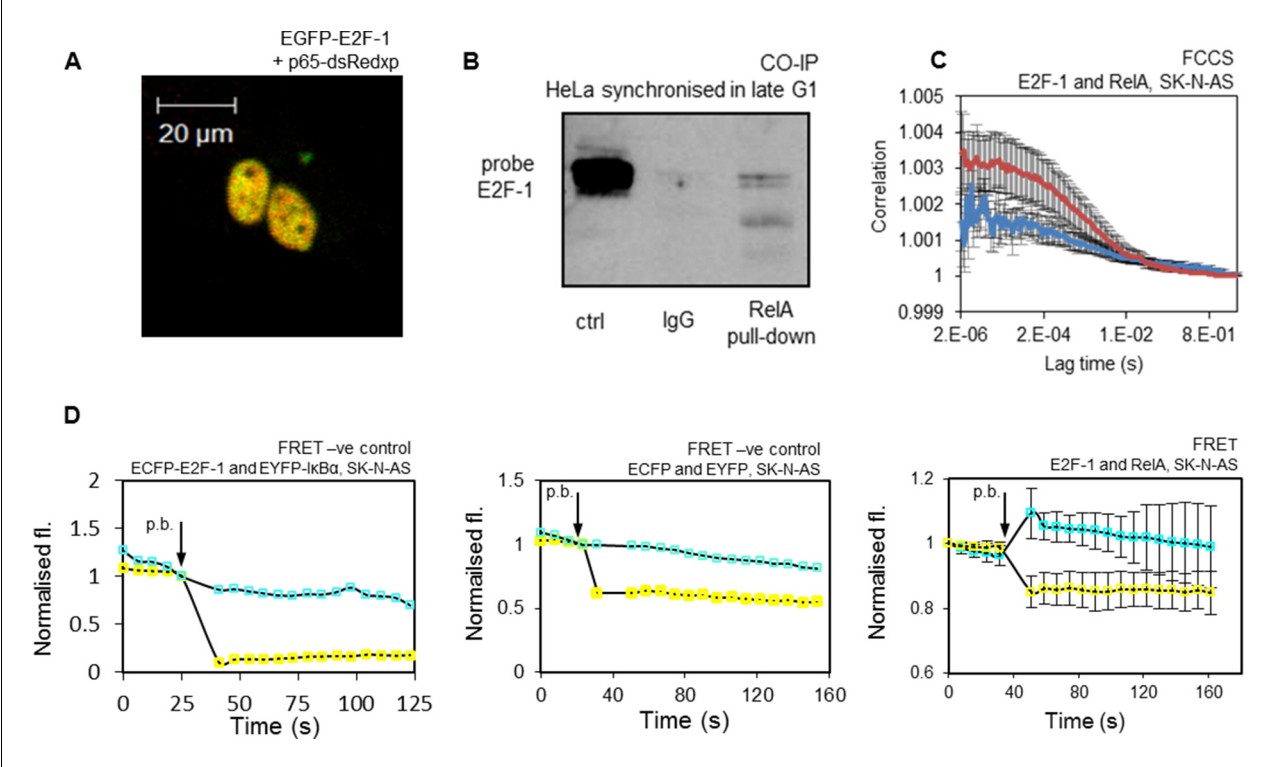

**Figure 5.** Interaction of E2F-1 with RelA. (**A**) Representative cell demonstrating co-localisation of E2F1-EGFP and RelA-dsRedxp upon transient transfection. (**B**) Co-Immunoprecipitation of E2F-1 with RelA pull down in HeLa cells synchronized in late G1 (HeLa cells used for this experiment due to their greater ease of synchronization). (**C**) FCCS assay between transiently transfected EGFP-E2F-1 and RelA-dsRedxp (red line) or empty-dsRedxp (blue line) fluorescent fusion proteins in single live SK-N-AS cells (+/- s.e.m based on 10 measurements from 10+ cells per condition). (**D**) Qualitative FRET assay between transiently transfected ECFP-E2F-1 and RelA-EYFP fluorescent fusion proteins in live SK-N-AS cells. First negative control between IkB-ECFP and EYFP-E2F1, and second negative control between free ECFP and EYFP fluorophores expressed in an SK-N-AS cell (shown are average ECFP and EYFP signals (+/- s.e.m based on 20 cells per condition normalised to pre-bleach intensity. p.b. indicates the time point at which photo-bleaching occurred).

When the initial mathematical model was used to simulate the effect of E2F-1 on the responsiveness of NF-κB to TNFα, the in silico simulations predicted that TNFα would induce immediate oscillations of free RelA (*Figure 6A*). In contrast, time-lapse live cell imaging of SK-N-AS cells stimulated with TNFα, showed that in cells expressing RelA-DsRedxp and EGFP-E2F-1 (which initially had nuclear RelA-DsRedxp), there was a delay before the onset of oscillations (*Figure 6B and D*). The length of this refractory period was on average ~4-fold longer than the peak1:peak2 timing in cells expressing RelA-DsRedxp alone (*Figure 6B* and [*Ashall et al., 2009*]). Altered model structures were investigated in order to resolve this discrepancy between experimental data and model predictions. One of the simplest altered models predicted that an E2F-1 target gene might stabilize IκBα (keeping NF-κB in the cytoplasm during S-phase [*Figure 6C*]). In support of this prediction, TNFα treatment of SK-N-AS cell populations transiently expressing EGFP-E2F-1 and RelA-DsRedxp led to reduced levels of phospho-S536-RelA and stabilized levels of IκBα (*Figure 6E*). Simulations of the response to TNFα from the revised model were consistent with the observed delay in oscillations in single cells expressing ectopic EGFP-E2F-1 (*Figure 6B and D*) and also with the inhibition or delay in the response during S-phase, but not during G1 or G2 (*Figure 2C*). Candidates for the E2F-1-regulated component(s) predicted by the revised model were therefore sought.

Previous studies had shown strong structural homology between E2F-1 and other E2F family members (*Tsantoulis and Gorgoulis, 2005*). E2F-4 is a transcriptional target of E2F-1 (*Xu et al., 2007*) and can be cytoplasmic during S-phase (*Lindeman et al., 1997*). E2F-4 (together with E2F family members) was therefore considered as a prospective candidate. We confirmed that ectopic expression of E2F-1 in cells resulted in increased E2F-4 expression, consistent with E2F-4 being a

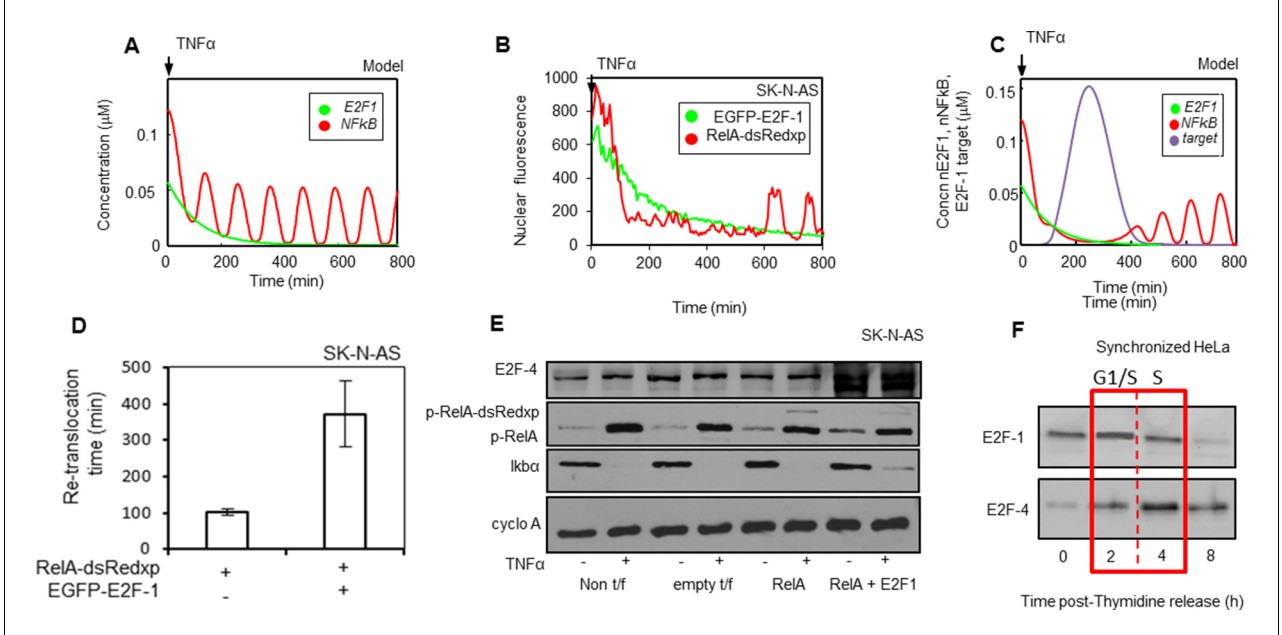

**Figure 6.** Mathematical modelling predicts an additional key component for NF-κB - cell cycle interactions: E2F-4 identified as a putative candidate. (A) Model simulations of RelA-dsRedxp dynamics when co-expressed with EGFP-E2F-1 in cells treated with TNFα. (B) Dynamics analysed in representative SK-N-AS cells treated with 10 ng/ml TNFα expressing RelA-dsRedxp and EGFP-E2F-1 (C) Model simulation of experimental conditions in B, incorporating interactions between NF-κB complexes and a putative E2F-1-induced target protein, subsequently proposed as E2F-4. (D) Analysis of average timing to second peak of NF-κB translocation following TNFα treatment in SK-N-AS cells expressing RelA-dsRedxp alone or with EGFP-E2F-1 (n=20 cells per condition, error bars show s.d.) (E) Assessment of the extent of RelA Ser[536] phosphorylation (p-RelA), E2F-4 and IκBα stability by western blot compared to cyclophilin A (cyclo A) amounts in SK-N-AS cells either untreated or treated with 10 ng/ml TNFα and expressing combinations of either untagged or fluorescent RelA-dsRedxp and EGFP-E2F-1. (F) Western blot of E2F-1 and E2F-4 in synchronized HeLa cells, where t=0 is late G1-phase.

transcriptional target of E2F-1 in these cells (*Figure 6E*). The profile of E2F-4 expression was found to be delayed relative to that of E2F-1 in the cell cycle, peaking in S-phase in synchronized HeLa cells (*Figure 6F*).

## E2F-4 and RelA physically and functionally interact

To further confirm the role of E2F-4 in the suppression of RelA translocation following TNFα treatment during S-phase, the physical and functional interactions between E2F-4 and RelA proteins in cells were investigated. When transiently expressed in either HeLa or SK-N-AS cells, both proteins were located in the cytoplasm (*Figure 7A*). Following TNFα treatment, the timing of RelA-DsRedxp translocation to the nucleus in both cell lines was delayed relative to the level of the fluorescent signal from EGFP-E2F-4 (*Figure 7B* for dynamic profiles and *Figure 7—figure supplement 1* for analysis). The physical interaction of endogenous E2F-4 and RelA proteins was supported by Co-IP from HeLa cells synchronized in S-phase (*Figure 7C*). No pull-down was observed in cells synchronised in late G1 phase (see *Appendix 1—figure 4*). This is the cell cycle stage when E2F-1, but not E2F-4 is at its peak expression level. This interaction was confirmed by acceptor photo bleaching FRET and FCCS data obtained from cells transiently expressing ECFP-E2F-4 and RelA-EYFP (for FRET) or RelA-dsRedxp and EGFP-E2F-4 (for FCCS) fluorescent fusion proteins (*Figure 7D and E* respectively). These data suggested that members of the E2F family have differing, but functionally linked, roles in the regulation of NF-κB dynamics. The observed dynamics could be represented by a mathematical model that recapitulates data (*Figure 6C*) from live cell imaging of the transient expression of the appropriate fluorescent fusion proteins (*Figure 5A* and *7A*, for details of modelling see Appendix Section B).

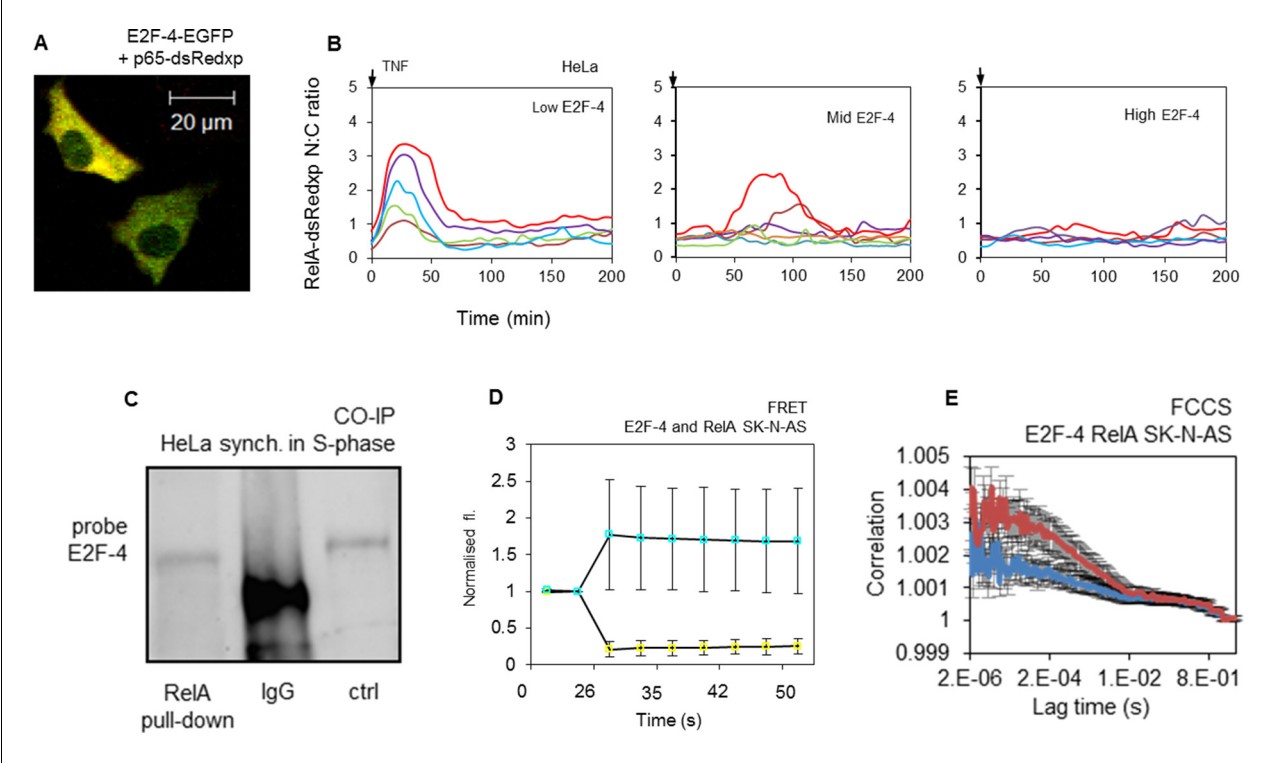

**Figure 7.** E2F-4 directly interacts with NF-κB and perturbs RelA dynamics in response to TNFα stimulation. (A) Single cell trajectories from groups of HeLa cells expressing RelA-dsRedxp and different levels of EGFP-E2F-4 showing the dynamics of RelA-dsRedxp after 10 ng/ml TNFα treatment (n=60 cells). (B) HeLa cells synchronized in S-phase, co-immunoprecipitated with anti-RelA antibody and probed for E2F-4. Also shown are IgG negative controls and whole cell lysate unsynchronized positive control (ctrl). (C) Representative SK-N-AS cells transiently transfected with RelA-dsRedxp and EGFP-E2F-4. (D) FRET assay in live SK-N-AS cells expressing ECFP-E2F-4 and RelA-EYFP fluorescent fusion proteins (shown are average ECFP and EYFP signals (+/- s.e.m) based on 20 cells per condition normalised to pre-bleach intensity. p.b. indicates the point of photo-bleaching. (E) FCCS assay in cells transiently expressing EGFP-E2F-4 and RelA-dsRedxp (red line) or dsRedxp (blue line) fluorescent proteins in single live SK-N-AS cells (+/- s.e.m based on 10 measurements in each of 10+ cells per condition).

The following figure supplement is available for figure 7:

**Figure supplement 1.** Analysis of RelA-dsRedxp dynamics in HeLa and SK-N-AS cells co-expressing EGFP-E2F-4 following TNFα stimulation.

## Analysis of the effect of the cell cycle on the NF-κB response at more physiological expression levels of E2F-1

The majority of experiments described above utilised transient expression of the E2F and RelA fusion proteins driven from a CMV promoter in a plasmid vector. Previous data had suggested that RelA fusion proteins expressed in a knock-in mouse are functional and fusion protein expression does not perturb the system (*De Lorenzi et al., 2009*). Our transcription analyses (*Figure 4*) suggested that E2F-1 N- and C-terminal fusion proteins also retained functional activity. However, as E2F proteins are normally expressed at specific stages of the cell cycle, ectopic expression from a strong constitutive promoter could give rise to out-of-context expression at inappropriate stages of the cell cycle (i.e. for E2F-1, stages other than late G1 and early S-phase). Therefore, expression of fusion proteins from these vectors might potentially show interactions that are not physiologically relevant. An additional complication in these experiments was that exogenous expression of E2F-1 (but not E2F-4) fluorescent fusion protein from a CMV promoter caused apoptosis when transfected alone. Interestingly this effect was rescued by co-expression with RelA.

To further validate the functional link between the E2F and RelA proteins, we sought to achieve more physiologically relevant levels and timing of the fluorescent fusion protein expression. To this end, stable HeLa cell lines were generated, with integrated Bacterial Artificial Chromosomes expressing E2F-1-Venus and RelA-DsRedxp under the control of their natural human gene

promotors and associated regulatory elements (see Appendix Section C). HeLa cells were chosen for this study based on their more consistent cell cycle timing (between cells) compared to SK-N-AS cells (as shown in *Figure 2—figure supplement 1*).

Stable cell lines were generated with a human E2F-1-Venus BAC construct, and showed the same pattern of synthesis and degradation of a transiently expressed FUCCI reporter for SCF (SKP-2) activity, indicating normal cell cycle progression (see *Figure 8—figure supplement 1*). All viable clones had relatively low expression of the E2F-1-Venus BAC, further suggesting that E2F-1 over-expression was detrimental to cell survival. Following the generation of these stable clones, a single clone was selected for the integration of a RelA-DsRedxp BAC into this cell line. This generated a dual stable clone of E2F-1-Venus and Rel-A-DsRedxp (termed C1-1). This clonal cell line showed a slight increase (~8%) in mean cell cycle length (with similar cell-to-cell variability) comparable with wild type HeLa (see *Appendix 1—figure 7*). Similar to wild type cells, TNFα treatment in the C1-1 cell line increased the variability in cell cycle timing compared to that of resting cells.

The slight change in mean cell cycle duration (~20 hr) in the dual BAC stable clonal cell line C1-1 was taken into account for inference of the dynamics of RelA-DsRedxp translocation at different cell cycle phases. The profile of E2F-1-Venus expression was used for assignment of the cell cycle stage at the time of stimulation cells based upon the time of peak E2F-1-Venus expression (*Figure 8—figure supplement 2*). This provided an alternative and faster method of virtual synchronisation to that used in *Figure 2*, allowing the assignment of G1, S and G2 phases to the data from the simulated BAC stable cells. The level of RelA translocation (*Figure 8B*) was then quantified for cells from each cell cycle phase. In agreement with data from the transiently transfected HeLa and SK-N-AS cells (*Figure 2*, *Figure 2—figure supplements 2* and *3*), the cells treated in late G1/S-phase showed higher amplitude RelA nuclear translocations, whereas Cells treated in S-phase showed a statistically significant suppression in S-phase RelA translocation compared to cells in early G1- or G2-phases (*Figure 8* and *Figure 8—figure supplement 3*).

Expression of the RelA-DsRedxp and E2F-1-Venus fusion proteins in the stable cell line was quantified through molecular counting of fluorophores via FCS (*Figure 8—figure supplement 4*). This gave an estimate of 310,000 ± 120,000 molcules of RelA-DsRedxp per cell. This figure was comparable to previous molecular estimates using FCS that had been obtained in stable cell lines generated using lentivirus (*Bagnall et al., 2015*), and previous estimates of RelA concentration using analytical chemistry (*Martone et al., 2003*; *Zhao et al. 2011*). RelA showed an approximate ratio of 3:1 ectopic to endogenous expression based on quantitative analysis of western blot data (see *Figure 8—figure supplement 4A*). By contrast, FCS analysis suggested that E2F-1-Venus expression was lower (24,000 ± 9100 molcules of E2F-1-Venus per cell). Western blot analysis (*Figure 8—figure supplement 4B*) suggested that there was an approximate ratio of 10:1 endogenous to ectopic levels). This might suggest selective pressure during cloning, as over-expression of E2F-1 has been reported to compromise cell viability (*Crosby and Almasan, 2004*). The apparent selective pressure against higher E2F-1 fusion protein expression was also in agreement with our own data that suggested that transient exogenous expression of E2F-1 fusion protein (but not E2F-4) alone caused apoptosis, but that this was rescued by co-expression of RelA. In the same manner observed with low EGFP-E2F1 expression from transient co-expression, the more physiological levels of E2F-1-Venus expression in the stably transfected cells suggested that RelA-DsRedxp remained predominantly cytoplasmic in unstimulated cells.

The interaction between E2F-1-Venus and RelA-DsRedxp following TNFα stimulation was measured by Fluorescence Cross-Correlation Spectroscopy (FCCS). A strong cross-correlation was confirmed in the nucleus (*Figure 8—figure supplement 4D*) indicating that the interaction uncovered by transient transfection with plasmids was not an artefact of over-expression, but was contextually relevant in relation to the cell cycle and RelA activation. Analysis of the dissociation constant (by FCCS) for the RelA-DsRedxp and E2F-1-Venus binding in the nucleus of TNFα–stimulated cells suggested a dissociation constant ($K_d$) of 12 nM (*Figure 8E*).

The stable and physiological co-expression of E2F-Venus and RelA-DsRedxp facilitated fluorescently labelled proteins to be observed over the course of a full cell cycle. Cells were virtually synchronized as previously described following stimulation with 10 ng/ml TNFα, and translocation of RelA-DsRedxp was plotted against the nuclear expression of E2F-1-Venus (*Figure 8—figure supplement 2*).

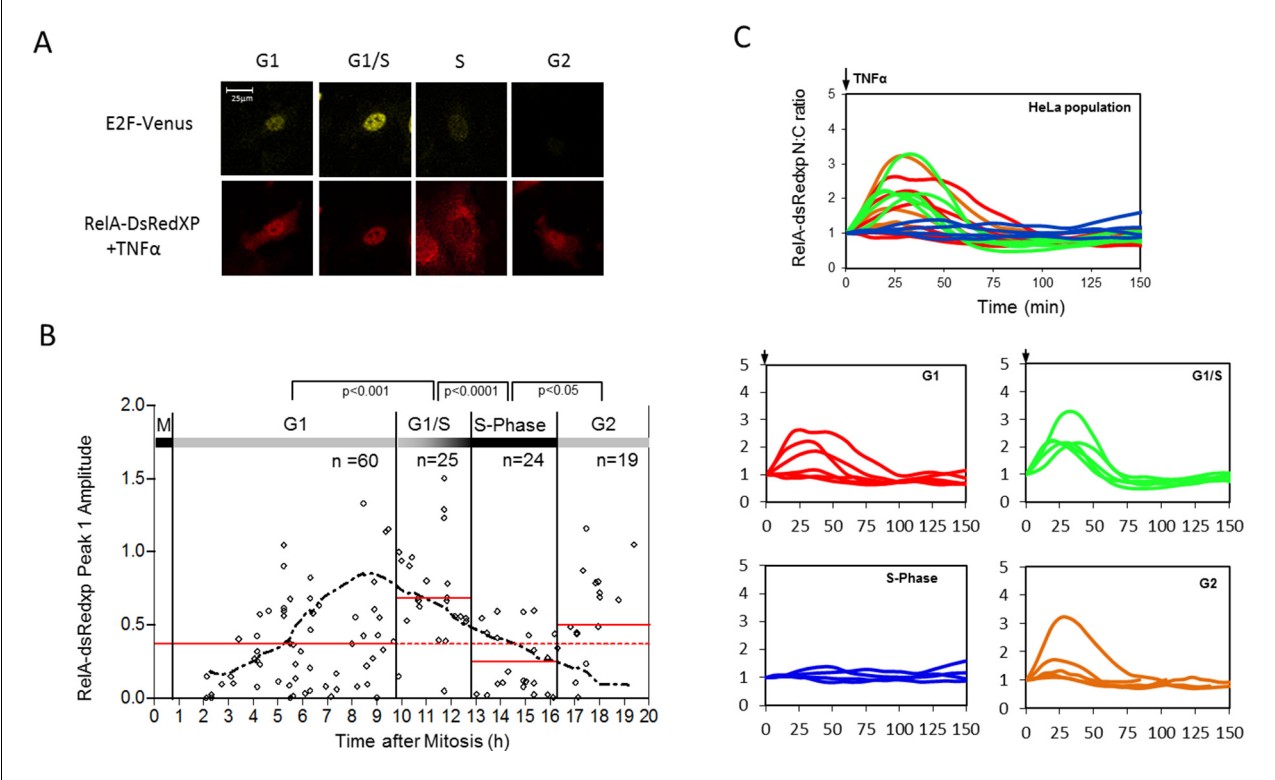

**Figure 8.** Effect of cell cycle timing on RelA-dsRedXP translocation in dual BAC HeLa cells (C1-1 line) that co-express E2F-1-Venus fusion protein. (**A**) Selected images from time-lapse experiment of dual BAC transfected HeLa stable clone 1-1 showing translocation of RelA-dsRedXP and E2F-1-Venus expression at different cell cycle phases. Cells were treated with 10 ng/ml TNFα. (**B**) Analysis of the dynamics of initial RelA-dsRedxp translocation in cells ordered at specific cell cycle times with respect to the peak of E2F-1 expression (n = 128). Data were analysed using nonparametric Anova analysis with Dunn correction for multiple comparisons. Red lines indicate mean normalised amplitude of NF-κB nuclear translocation for different cell cycle phases, and the population average (dotted red line). Analysis of nuclear RelA occupancy was assessed in virtually synchronised C 1-1 cells, based on time from cell division and relative to peak E2F-1-Venus expression level. RelA-dsRedxp localization was visualized to allow quantification of translocation, following treatment with 10 ng/ml TNFα. The dotted black line shows the spline fitted level of E2F1 at different times and cell cycle stages (see also *Figure 8—figure supplement 1* below). Statistical analysis showed a difference between G1 vs S, and G2 vs S with respect to distribution of amplitude of the RelA translocation response. (**C**) RelA-dsRedxp dynamics following 10 ng/ml TNFα treatment in asynchronous cells (left panel) and cells virtually synchronised into G1, G1/S, S and G2 phases. The data for each cell was normalised to the amplitude (N:C ratio) at t = 0 min.

The following figure supplements are available for figure 8:

**Figure supplement 1.** Virtually synchronized HeLa C 1-1 cells.

**Figure supplement 2.** Physiological and functional expression of E2F-1-Venus in stable BAC-transduced HeLa cells.

**Figure supplement 3.** Analysis of the expression of E2F-1-Venus and RelA-DsRedxp translocation in single C1-1 HeLa cells stimulated with 10ng/ml TNFα at different cell cycle phases.

**Figure supplement 4.** Expression and interaction of RelA-dsRedxp and E2F-1-Venus.

We also investigated the consequences of knocking down both E2F-1 and E2F-4 using siRNA. Imaging experiments showed E2F-1 knockdown did not prevent cell cycle progression, and did not affect the heterogeneity of population response upon TNFα stimulation (data not shown), perhaps indicating compensation by other E2F family members. In addition, our mathematical model predicted that knocking down E2F-1 might not substantially affect the repression of the NF-κB response in S-phase, which was instead predicted to be due to the effect of E2F-4 expression. However, knock-down of E2F-4 was found to be lethal to cells (*Crosby and Almasan, 2004*) preventing time lapse analysis. A key additional consideration is the overlapping roles of other E2F family members,

which makes knock-down of individual E2F proteins unpredictable, due to potentially co-operative and/or redundant functions.

## Discussion

Biological timing plays a key role in the encoding and decoding of biological information. Of particular interest is the role of biological oscillators, which can have very different cycle periods. A key question is how they may interact to robustly control essential biological processes. Here, we propose a reciprocal relationship between two oscillators, NF-κB signalling and the cell cycle.

TNFα stimulation in S-phase showed a suppressed and delayed translocation of RelA, with no observable perturbation to cell cycle timing. In contrast, stimulation in late G1 showed strong translocation of RelA (*Figure 2*) and led to significant lengthening of the cell cycle (*Figure 3*). These data suggest that cells use the G1/S checkpoint to prioritize between inflammatory signalling and the onset of DNA replication prior to cell division (see schematic diagram in *Figure 9*). The presence of a mechanism for prioritization between the important processes of cell proliferation and inflammation suggests that an inflammatory response during DNA replication might be detrimental to the cell.

The data showing that TNFα stimulation alters cell cycle timing in a cell cycle phase-dependent manner is intriguing (*Figure 3*). However, our data do not identify a specific mechanism by which TNFα may regulate cell cycle length. The observation that the effect of TNFα stimulation on cell cycle lengthening appears to be specific to G1/S- rather than S-phase suggests that this may occur by delaying transition through the G1/S checkpoint. One hypothesis is that this might occur through NF-κB modulation of E2F family transcriptional activity. At the same time, the system is more complex as NF-κB is known to regulate the expression of other key cell cycle regulating proteins. Important examples include Cyclin D (*Guttridge et al., 1999*; *Hinz et al., 1999*, *Sée et al., 2004*), and p21$^{waf1/cip1}$ (*Basile et al. 2003*). Therefore, there is undoubtedly a more complex set of interactions between NF-κB and the control of cell proliferation and cancer (*Perkins and Gilmore, 2006*).

As well as a number of studies that suggest a physical interaction between E2F and NF-κB proteins (*Kundu et al., 1997*; *Chen et al., 2002*; *Tanaka et al., 2002*, *Shaw et al., 2008*; *Palomer et al., 2011*), there have been a few previous studies that have suggested that this interaction might have functional importance. Araki *et al.* described an NF-κB-dependent mechanism for growth arrest mediated by a dual mechanism. They suggested that E2F-1-dependent transcription was inhibited by IKK activation and that E2F-4 was phosphorylated directly by IKK resulting in increased activity of the E2F-4/p130 repressor complex (*Araki et al., 2008*). Their study did not assume direct interactions between the E2F and Rel proteins and did not take into account protein dynamics. Nevertheless, their conclusions are very complementary to the present study.

Another study by Tanaka et al. focused on the combined role of E2F-1 and c-MYC in the inhibition of NF-κB activity (*Tanaka et al., 2002*). This study demonstrated interactions between E2F-1 and both RelA and p50. Rather than focusing on cell division, their study showed that inhibition of RelA activity by E2F-1 resulted in increased apoptosis. Since both the NF-κB and E2F families of transcription factors have important roles in the control of apoptosis (*Phillips and Vousden, 2001*; *Kucharczak et al., 2003*; *Crosby and Almasan, 2004*), it is therefore interesting to speculate that the levels of different E2F proteins at different cell cycle stages may regulate cell fate decision making in collaboration with signalling systems such as NF-κB.

One important conclusion of the current study is the physical interaction of RelA with E2F-1 and E2F-4 proteins. It is however not necessary to assume strong binding and sequestration into different cellular compartments. Instead, control of cross-talk could be a consequence of mutual control of gene expression. We provide some data that suggests that E2F-1 and IκBα may compete for binding to RelA (see *Figure 4E*). We suggest that control may be achieved through repression of the IκBα feedback loop (and perhaps other negative feedbacks, such as A20). However, it might be that other genes are differentially activated through the combined action of these transcription factors. In support of this, Garber *et al.* performed a study in dendritic cells where they studied a panel of transcription factors by ChIP-Seq following LPS stimulation. Their data suggested that E2F-1 and RelA are common transcription factor pairs that were bound together at a large set of functionally important gene promoters (see data in *Figure 3B* of *Garber et al., 2012*). It therefore seems likely

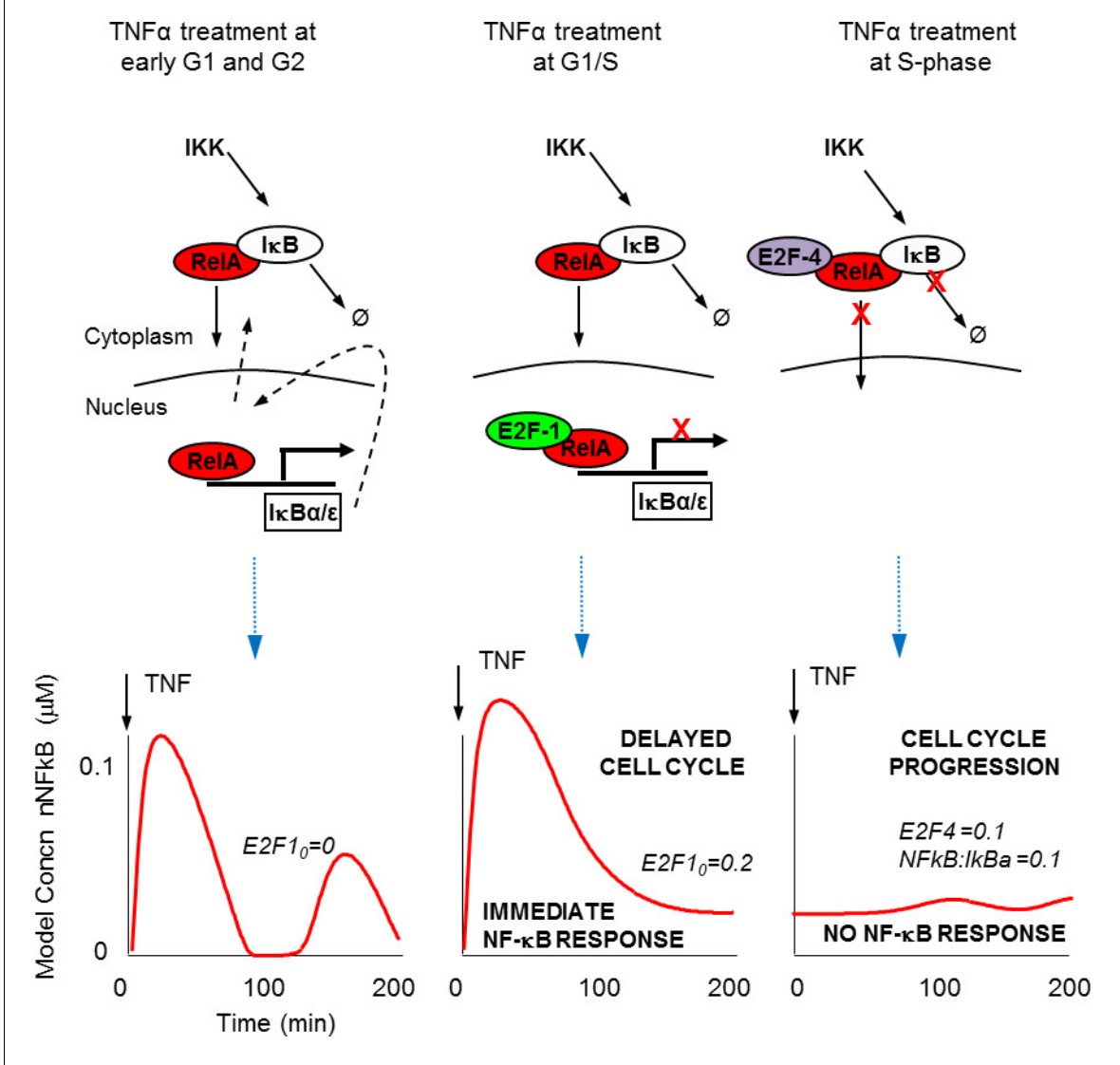

**Figure 9.** Schematic representation of NF-κB – E2F interactions. (**A**) Predicted mechanisms for NF-κB interaction with E2F proteins over the G1/S transition (**B**) Model simulations of single cell behaviour.

that these proteins mutually regulate patterns of transcriptional activity, controlling the expression of downstream feedback genes, cell proliferation and apoptosis.

We describe a mechanism for E2F-1 that suggests competition with IκBα for NF-κB binding. This was effectively described by the model (see also *Figure 9*), and was used to predict the role for an E2F-1 target gene, upregulated in S-phase. Our data support E2F-4 as a candidate for this E2F-1 target gene. It should be noted however, that the E2F family of proteins may all play a role in this complex system. A surprising characteristic of E2F-4 is its predominantly cytoplasmic localisation in some cell types. As a result, we were unable to perform a competition localisation experiment (as for E2F-1, *Figure 4E*). We cannot therefore comment on whether E2F-4 also competes with IκBα for RelA binding. Therefore, the model (both mathematical model and schematic model in *Figure 9*) encode E2F-4 binding as a ternary complex to RelA and IκBα together. We stress that this is only one possible mechanism, but we have used this formulation since it is the simplest model that is consistent with all of our data. As described by Araki *et al.* (see above) there may be other components involved such as IKK-mediated E2F-4 phosphorylation (*Araki et al., 2003*).

Functional and context-dependent coupling between dynamic cellular processes (such as the cell cycle, the circadian clock [*Yang et al., 2010*; *Bieler et al., 2014*; *El Cheikh et al., 2014*], or p53 [*Toettcher et al., 2009*]) is emerging as a common theme in intracellular signalling (*Ankers et al., 2008*; *White and Spiller, 2009*; *Spiller et al., 2010*). The present study has characterized a dynamic and functional interaction between NF-κB and the cell cycle systems, which each oscillate with different periods. Coupling between cellular processes (e.g. at the G1/S commitment point) can have contrasting effects on cell fate. Such temporal communication between processes represents a way for cells to gate their biological signals and coordinate and prioritize cell fate decisions in response to changes in their environment. In a wider context, understanding how (and when) these dynamic interactions occur could yield important therapeutic targets for fields such as cancer chronotherapy (*Choong et al., 2009*; *Lévi et al., 2010*).

## Materials and methods

### Materials
Human recombinant TNFα was supplied by Calbiochem (UK). Tissue culture medium was supplied by Invitrogen (UK) and Fetal Bovine Serum (FBS) was from Harlan Seralab (UK). All other chemicals were supplied by Sigma (UK) unless stated otherwise.

### Plasmids
All plasmids were propagated using *E. coli* DH5α and purified using Qiagen Maxiprep kits (Qiagen, UK). NF-κB-Luc (Stratagene, UK) contains five repeats of an NF-κB-sensitive enhancer element upstream of the TATA box, controlling expression of luciferase. Luciferase reporter CyclinE-Luc was obtained from Peggy Farnham (University of Wisconsin-Madison, USA). EGFP-E2F-1 and EGFP-E2F-4 contain the Enhanced Green Fluorescent Protein (EGFP) gene (Invitrogen, UK) fused to the N-terminal ends of the human E2F-1 and E2F-4 gene fragments (kind gifts from Emmanuelle Trinh, BRIC, Denmark). Similarly, ECFP-E2F-1 and ECFP-E2F-4 contain the Enhanced Cyan Fluorescent Protein (ECFP) gene (Invitrogen, UK) RelA-DsRedxp contain the optimised DsRed Express protein (DsRedxp) gene (Clontech, CA) fused to the c-terminal end of human RelA gene (described previously in *Nelson et al. (2002)*. RelA-EYFP contain Enhanced Yellow Fluorescent protein (EYFP) gene (Invitogen, UK) fused to the C-terminal end of human RelA gene.

### Cell culture
SK-N-AS neuroblastoma (cat.no. 94092302) and HeLa cervical carcinoma (Cat. No. 93021013) cell lines were obtained from European Collection of Authenticated Cell Cultures (ECACC). Cells were cultured and frozen down to form a low passage working stock. Subsequent working stocks were used for no more than 10 passages. Working stocks were screened to ensure the absence of mycoplasma every 3 months using LookOut Mycoplasma PCR Detection Kit (Cat. No. D9307 Sigma, UK). For confocal fluorescence microscopy and immuno-cytochemistry, SK-N-AS and HeLa cells were plated on 35 mm glass-bottom dishes (Iwaki, Japan and Greiner, Germany) at $1\times10^5$ cells per dish in 3 ml medium. HeLa cells were plated at $5\times10^4$ cells per dish in 3 ml medium. 24 hr post-plating, the cells were transfected with the appropriate plasmid(s) using Fugene 6 (Boehringer Mannheim/Roche, Germany). The optimized ratio of DNA:Fugene 6 used for transfection of HeLa or SK-N-AS cells was 2 µg DNA with 4 µl Fugene 6 and 0.8 µg DNA with 1.2 µl Fugene 6 respectively.

For Co-IP assays, SK-N-AS cells were plated on 100 mm tissue culture dishes (Corning, USA) at $4.5\times10^6$ cells per dish in 10 ml medium. For western blotting, semi-quantitative and quantitative PCR, HeLa and SK-N-AS cells were plated on 60 mm tissue culture dishes (Corning, USA) at $5\times10^5$ and $1\times10^6$ cells respectively per dish in 5 ml medium.

### G1/S Cell cycle synchronisation via double Thymidine block
24 hr post-plating, 2 mM Thymidine was added to the culture medium. Following a 19 hr incubation, cells were washed and fresh medium added. Following a 9 hr incubation, 2 mM Thymidine was again added to the culture medium and the cells incubated for a further 16 hr. Cells were then washed and fresh media added. Following release from Thymidine block, the G1/S-synchronized cells were either imaged or incubated (at 37°C, 5% $CO_2$) for the indicated duration prior to cell lysis or fixation.

## Treatment of cells with TNFα

For confocal fluorescence microscopy, the cells were treated in-situ between imaging acquisitions after an indicated pre-treatment incubation period (usually 24 hr post-transfection). For western blotting and q-PCR experiments, the cells were treated with TNFα 24 hr post-plating. The cells were imaged either immediately after treatment, or incubated (at 37°C, 5% $CO_2$) for the indicated duration prior to cell lysis or fixation.

## Fluorescence microscopy

Confocal microscopy was carried out as described (*Nelson et al., 2004*) using either 20x Fluar 0.8 NA or 63x Planapochromat 1.4 NA objectives. CellTracker (*Shen et al., 2006*; *Du et al., 2010*) was used for data extraction. For RelA fusion proteins, mean fluorescence intensities were calculated for each time point for both nucleus and cytoplasm then nuclear:cytoplasmic (N:C) fluorescence intensity ratios were determined. For time lapse microscopy, a modified version of the Autofocus macro (an improved version of the Autotimeseries macro [*Rabut and Ellenberg, 2004*]) was used.

## Analysis of cell cycle progression

The cell cycle duration and G1/S timing of SK-N-AS and HeLa cells was analysed using live-cell imaging of successive cell divisions to determine typical cell cycle duration. In addition, the cell cycle dynamics were quantified expressing Fluorescence Ubiquitin-based Cell Cycle Indicators (FUCCI, [*Sakaue-Sawano et al., 2008*]) (*Figure 2—figure supplement 1*). The crossing point in fluorescent levels from FUCCI markers of APC and SCF E3 ubiqutin ligase was used as an indication of G1/S transition in the cells (*Figure 2—figure supplement 1B*). Mitosis to mitosis timings were determined in non-transfected cells, as well as in cells transfected with RelA-dsRedXP and the dual BAC cell line (*Appendix 1—figure 7*) For the BAC cell line that expressed E2F-1-Venus it was only possible to use the single SCF FUCCI G1 vector (due to fluorescent protein spectral overlap).

## Virtual synchronization

Cells were imaged for ~30 hr prior to TNFα treatment in order to capture each cell passing through mitosis. The timing of TNFα treatment relative to mitosis for each cell was then calculated. Events following TNFα treatment (i.e. the dynamics of RelA-DsRedxp translocation, or cell cycle duration) could then be correlated to inferred cell cycle phase at the point of treatment. Dual BAC cell lines were imaged for an entire cell cycle. Cells were aligned based upon normalised peak amplitude of E2F-1-Venus, and virtually synchronised based upon alignment of peak E2F-1 expression and the relative timing of TNFα stimulation. Cell cycle boundaries were inferred through characterization of cell cycle progression through transfection of FUCCI G1 phase marker construct (*Figure 2—figure supplement 1*).

## Flow cytometric DNA analysis

HeLa cells were cultured in 100 mm dishes. Following trypsinization, and resuspension in 1 ml of medium the cells were stained by addition of 250 µl of 50 µg/ml propidium iodide, 0.15% TritonX-100, and 150 µg/ml RNase A before analysis in an Altra flow cytometer (Beckman Coulter).

## Förster resonance energy transfer (FRET) microscopy

FRET was carried out using a Zeiss LSM510 with 'META' spectral detector mounted on an Axiovert 100S microscope with a 63x Planapochromat, 1.4 NA oil-immersion objective (Zeiss). ECFP and EYFP (*Karpova et al., 2003*) were excited with 458 nm laser light, emitted fluorescence was collected in 8 images each separated by 10 nm between 467 nm and 638 nm in lambda scanning mode. Separation of ECFP and EYFP fluorescence spectra was carried out using the linear unmixing algorithms of the Zeiss LSM510 software (Zeiss), using reference spectra taken from cells expressing the ECFP or EYFP fusion proteins alone or untransfected cells. The fluorescence spectrum was separated into ECFP, EYFP and background signals. FRET was assayed by acceptor (EYFP) photo-bleaching. Bleaching was accomplished using 50 iterations of 514 nm laser light with no attenuation from the acousto-optical tuneable filter (AOTF).

## Fluorescence correlation spectroscopy (FCS) and fluorescence cross-correlation spectroscopy (FCCS)

FCS and FCCS was carried using either a Zeiss LSM780 or Zeiss 710 with Confocor 3 mounted on an AxioObserver Z1 microscope with a 63x C-apochromat, 1.2 NA water-immersion objective. Zen 2010B software was used for data collection and analysis. EGFP fluorescence was excited with 488nm laser light and emission collected between 500 and 530 nm. DsRed-express was excited with 561nm laser light and emission collected between 580 and 630 nm. The protocols as outlined in Kim et al. (*Kim et al., 2007*) were followed, with 10 x 10 s runs used for each measurement. FCS was used to quantify the total number of fluorescent molecules per cell as previously described (*Bagnall et al., 2015*). The confocal volume had previously been estimated at 0.59 ± 11 fL (mean ± SD) using Rhodamine 6G of known diffusion rate, and WT HeLa cells in suspension were imaged by confocal microscopy to give volume estimates of 1420 ± 490 fL and 6110 ± 3580 fL for nucleus and cytoplasm respectively. (For FCCS controls see Appendix Section E).

## Co-immunoprecipitation

HeLa cells synchronized at G1/S or S-phase were washed with room temperature PBS and lysed with modified RIPA buffer (50 mMTris-HCl pH 7.4, 150 mM NaCl, 1 mM EDTA, 1% NP-40) including a 1:100 dilution of Protease Inhibitor cocktail (Sigma, UK), PMSF and phosphatase inhibitor (Phos Stop, Roche). Immunoprecipitation was carried out using Immunoprecipitation kit-Dynabeads Protein G (Invitrogen) with anti-RelA antibody (#3034, Cell Signaling, MA, USA). The samples were analyzed by western blotting using anti-E2F-1(Cell Signaling, #3742) or anti E2F-4 (Santa Cruz, C-20 sc-866) antibodies.

## q-PCR

The RNeasy Mini Kit (Invitrogen, UK) was used to extract mRNA from the cells following manufacturer's instructions, using the primers: IκBα left TGGTGTCCTTGGGTGCTGAT right GGCAG TCCGGCCATTACA, IκBε left GGACCCTGAAACACCGTTGT right CCCCAGTGGCTCAGTTCAGA, E2F-1 left TGCAGAGCAGATGGTTATGG right TATGGTGGCAGAGTCAGTGG, cyclophilin A left GCTTTGGGTCCAGGAATG right GTTGTCCACAGTCAGCAATGGT.

## Luciferase reporter assay

Luciferase reporter assay were carried out as described in *White et al. (1990)*, using a LUMIstar plate reading luminometer (BMG, Germany).

## Immuno-cytochemistry (ICC)

HeLa cells were prepared using combinations of the above techniques, typically involving synchronization and/or TNFα stimulation of cells seeded at appropriate density into 35 mm glass-bottomed dishes. Dishes were subsequently washed three times with PBS and fixed with 1 ml 4% paraformaldehyde for 15 min. Dishes were then washed three times with PBS, and 'blocked' to prevent non-specific antibody binding with the addition of 1–2 ml of 1% BSA, 0.1% Triton X-100 (in PBS) from 20 min up to overnight. The primary antibody (or antibodies for dual-staining), dissolved in Ab Buffer (1% BSA, 0.1% Triton X-100 in PBS), were added to the dishes for 60/90 min at a 1:2000 dilution. Dishes were then washed (3x1 ml) with Ab buffer for 10 min. Secondary Antibody(s) were subsequently added to the dishes (Cy3-anti-mouse, 1:200 dilution (Sigma), FITC Rabbit, 1:200 [AbCam]) for 30/45 min respectively, prior to 3 sequential washes of PBS blocking buffer (described above). Following the addition of fluorescent secondary antibodies, dishes were covered in aluminium foil and left in 2 ml PBS prior to imaging.

## Western blotting

Whole cell lysates were prepared at the indicated times after stimulation. Membranes were probed using the following antibodies: anti-IκBα (#9242, Cell Signaling, MA), anti-RelA (#3034, Cell Signaling, MA), anti-phospho-RelA (Ser 536) (#3031, Cell Signaling, MA), anti-IκBα (#9242, Cell Signaling, MA), anti-E2F-1 (#KH-95, Millipore Biotechnology, USA), anti-E2F-4 (sc-866, Santa Cruz), α-Tubulin Antibody (#2144 Cell Signaling, MA), and anti-cyclophilin A (#07–313, Millipore Biotechnology, USA).

## Acknowledgements

We thank Rob Beynon, David Rand, Daniel Larson, and Neil Perkins for critical reading of the manuscript and Bela Novak for helpful discussions. We also thank Glyn Nelson and James Johnson for advice at the outset of the project, and Jan Harries for E2F-4 cloning work. The HA-E2F-1 and E2F-4 expression vectors were kind gifts from Dr E Trinh, and K Helin (Copenhagen, Denmark). The work was supported by BBSRC grants BBH0137252, BBF0059382/BBF00561X1 and BB/K003097/1; MRC grants G0500346 and MR/K015885/1; Professor John Glover Memorial Fellowship to CVH; BBSRC David Phillips Fellowships to VS (BBC5204711) and Pawel Paszek (BBI0179761) and a BBSRC studentship to JB (BBF5290031). Carl Zeiss Ltd provided training and technical support.

## Additional information

### Funding

| Funder | Grant reference number | Author |
| --- | --- | --- |
| Biotechnology and Biological Sciences Research Council | BBH0137252 | John M Ankers<br>David G Spiller<br>Violaine Sée<br>Michael RH White |
| Biotechnology and Biological Sciences Research Council | BBF5290031 | James Boyd<br>David G Spiller<br>Dean A Jackson<br>Michael RH White |
| Medical Research Council | G0500346 | Sheila Ryan<br>David G Spiller<br>Pawel Paszek<br>Violaine Sée<br>Michael RH White |
| Professor John Glover Memorial Fellowship | N/A | Claire V Harper |
| Biotechnology and Biological Sciences Research Council | BB/K003097/1 | Claire V Harper<br>David G Spiller<br>Dean A Jackson<br>Pawel Paszek<br>Michael RH White |
| Medical Research Council | MR/K015885/1 | David G Spiller<br>Dean A Jackson<br>Michael RH White |
| Biotechnology and Biological Sciences Research Council | BBF0059382/BBF00561X1 | David G Spiller<br>Dean A Jackson<br>Violaine Sée<br>Michael RH White |
| Biotechnology and Biological Sciences Research Council | BBI0179761 | Pawel Paszek |
| Biotechnology and Biological Sciences Research Council | BBC5204711 | Violaine Sée |

The funders had no role in study design, data collection and interpretation, or the decision to submit the work for publication.

### Author contributions

JMA, Designed and performed the experiments and modelling and wrote the paper, Analysis and interpretation of data, Contributed unpublished essential data or reagents; RA, Performed co-IP experiments, Made and analysed experiments with the E2F-1 BAC and cell line; NAJ, Carried out western blotting, Imaging, siRNA knockdown experiments, Assisted with FCCS experiments on BAC stable cell lines, Wrote the manuscript, Analysis and interpretation of data, Contributed unpublished essential data or reagents; JB, FCCS experiments and edited the paper, Acquisition of data, Analysis and interpretation of data, Drafting or revising the article; SR, Performed Western blotting and Q-PCR and helped with FRET experiments, Analysis and interpretation of data; ADA, Led the BAC

design and cloning strategy, Contributed unpublished essential data or reagents; CVH, Provided advice at the outset of the project and assisted with qPCR, Acquisition of data; LB, Assisted with mathematical model analysis, Analysis and interpretation of data; DGS, Provided expert help with all imaging experiments and wrote the paper, Conception and design, Acquisition of data, Analysis and interpretation of data, Contributed unpublished essential data or reagents; DAJ, Assisted with BAC development and advised on and edited the paper, Drafting or revising the article; PP, Expert advice on mathematical model building and edited the paper, Analysis and interpretation of data, Drafting or revising the article; VS, Provided expert help on E2F cloning and IP experiments, Advised on project planning and edited the paper, Drafting or revising the article, Contributed unpublished essential data or reagents; MRHW, Directed the project and wrote the paper, Conception and design, Analysis and interpretation of data

## Author ORCIDs
Dean A Jackson, http://orcid.org/0000-0002-1570-5287
Michael RH White, http://orcid.org/0000-0002-3617-3232

## Additional files

### Major datasets

The following dataset was generated:

| Author(s) | Year | Dataset title | Dataset URL | Database, license, and accessibility information |
|-----------|------|---------------|-------------|-------------------------------------------------|
| Ankers JM, Awais R, Jones NA, Boyd J, Ryan S, Adamson AD, Harper CV, Bridge L, David G Spiller, Dean A Jackson, Pawel Paszek, Violaine Sée, Michael RH White | 2015 | Data from: Dynamic NF-κB and E2F interactions control the priority and timing of inflammatory signalling and cell proliferation. | https://dx.doi.10.5061/dryad.th18q | Available at Dryad Digital Repository under a CC0 Public Domain Dedication |

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

# Appendix

## Section A: Thymidine synchronisation and analysis

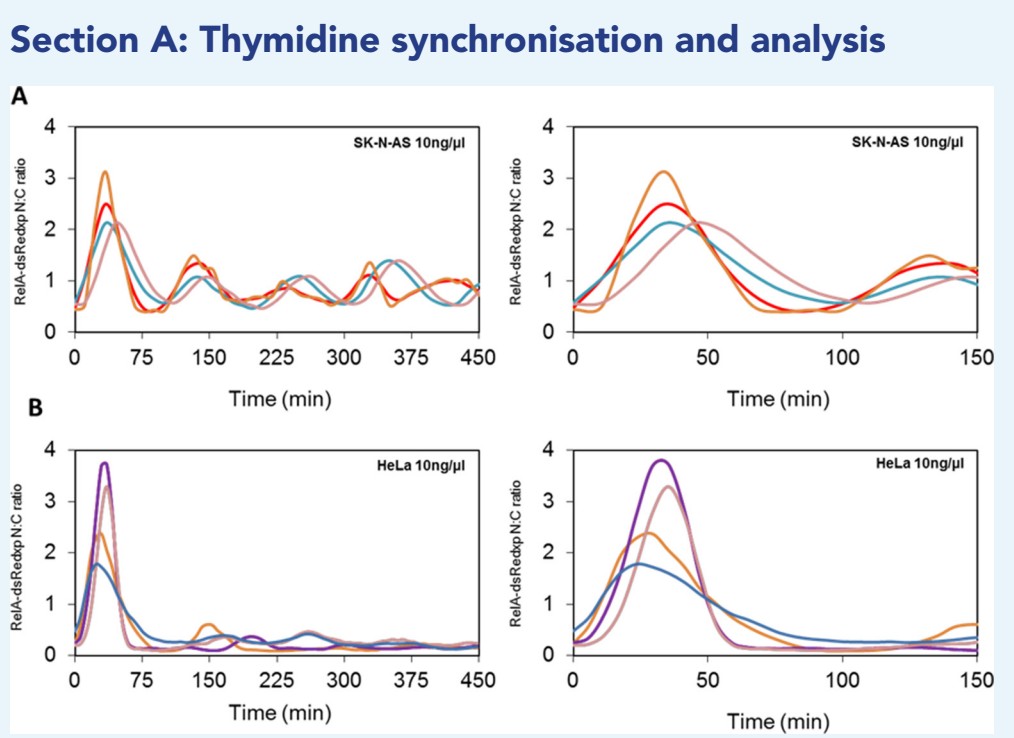

**Appendix 1—figure 1.** Oscillations in the NF-κB system. (**A**) Dynamics of RelA-dsRedxp in transiently transfected SK-N-AS cells following 10 ng/µl TNFα stimulation, plotted over 450 and 150 min respectively. (**B**) Dynamics of RelA-dsRedxp in transiently transfected HeLa cells following 10 ng/µl TNFα stimulation, plotted over 450 and 150 min respectively.

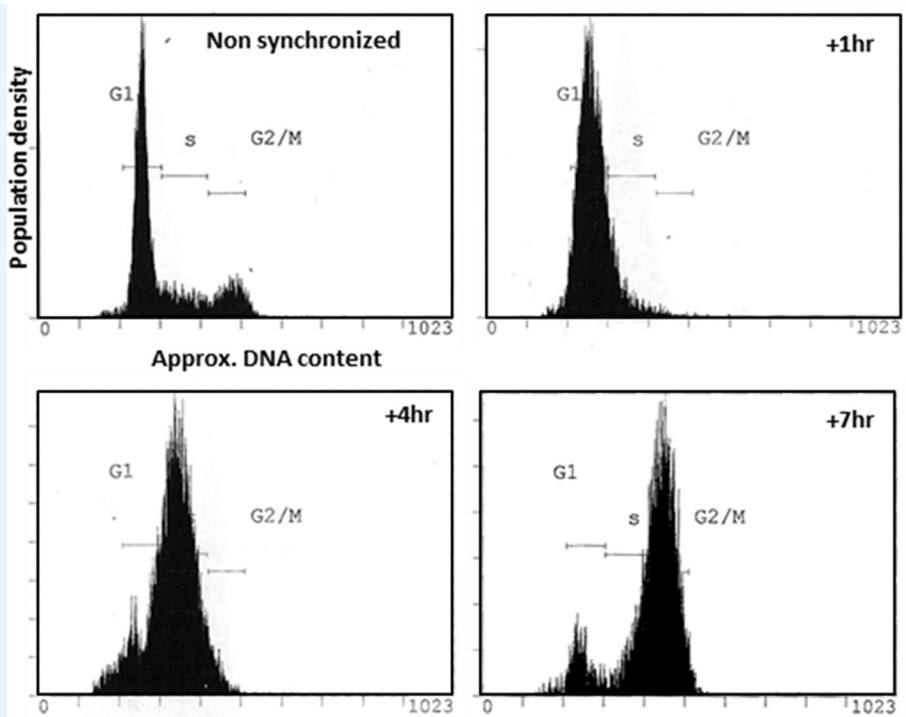

**Appendix 1—figure 2.** Use of double-Thymidine block to synchronize HeLa cells at G1/S. Flow cytometric analysis of the distribution of DNA content of non-synchronized HeLa cells and cells harvested at relevant times post-release from Thymidine block.

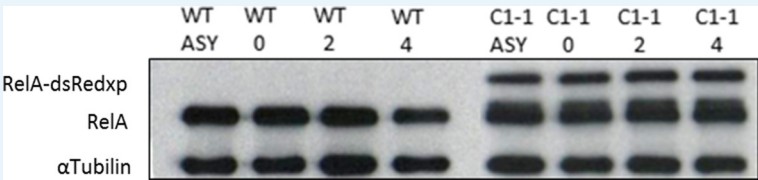

**Appendix 1—figure 3.** Effect of Double-Thymidine block on tagged and endogenous RelA levels HeLa cells. Endogenous RelA and tagged RelA-DsRedxp expression levels in unsynchronised and synchronised WT-HeLa and double BAC stable cells. Synchronized fractions at 0, 2 and 4 hr post release of thymidine block. α-Tubulin used as loading control.

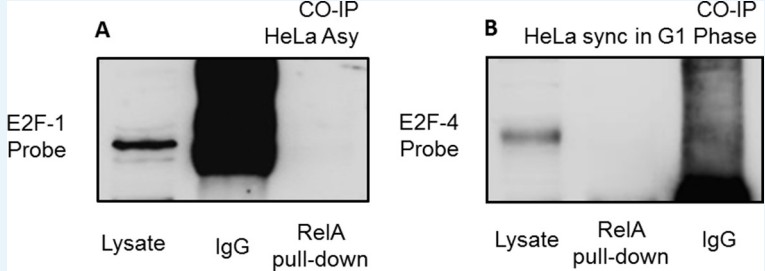

**Appendix 1—figure 4.** Negative Co-IP. (**A**) Co-Immunoprecipitation of E2F-1 with RelA (pulled down with a RelA antibody) in asynchronous HeLa cells showing no detectable band of E2F-1. (**B**) Co-Immunoprecipitation of E2F-4 with RelA (pulled down with a RelA antibody) in HeLa cells synchronized in late G1-phase cells showing no detectable band of E2F-4.

## Section B: The NF-κB:E2F mathematical models

All modelling work was implemented using MATLAB2010 (MathWorks, USA), and simulated using MATLAB ordinary differential equation solver ODE15s. Analysis of simulated time course data was performed in both Microsoft Excel and MATLAB.

Model equations and parameters are shown in SI *Appendix 1—table 2* and *3* respectively. The deterministic model of TNFα-induced NF-κB signalling (*Ashall et al. 2009*), consisting of a system of ordinary differential equations for species concentrations with respect to time, was extended in two steps. Firstly, the physical interaction between E2F-1 and RelA was included. It was assumed that E2F-1 competes with IκBα for binding of free NF-κB with similar affinity. In addition, free IκBα actively disrupted the NF-κB:E2F-1 complex, while IκBα:NF-κB complex was unaffected by free E2F-1. Secondly, the physical interaction between cytoplasmic NF-κB complexes and E2F-4 was included. E2F-4 was modelled as an E2F-1 responsive gene which, upon translation, forms complexes with NF-κB and NF-κB:IκBα, which were not targeted by IKK. This reduced the system sensitivity to TNFα treatment for a prolonged period.

A typical simulation experiment involved three sequential stages: equilibration, transfection and TNFα treatment (shown below):

1. The model was initialized by setting neutral IKK and cytoplasmic IκBα:NF-κB to 0.1 μM and other variables to 0. The system was then equilibrated for 1000 min to reach the untreated steady state.

2. Initial conditions from the end of the equilibrium stage were amended to mimic cell transfection. For example, in the case of E2F-1 and RelA co-transfection 0.1 μM cytoplasmic protein was added to the respective initial conditions.

3. The equilibrated and transfected model was simulated for 800 min (with TR set to either 1 or 0 depending on whether treatment was simulated or not).

Simulations of the full model are shown below. Simulation protocols used throughout the manuscripts are summarized below.

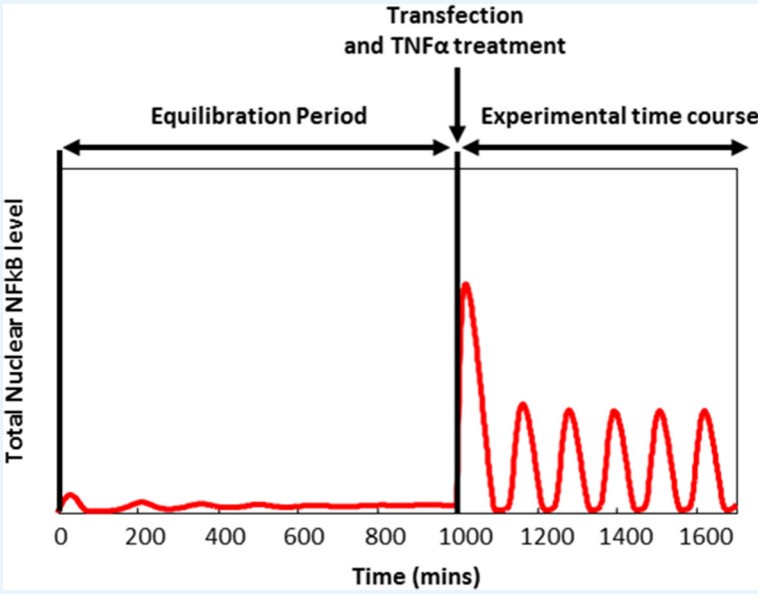

**Appendix 1—figure 5.** A typical simulation protocol of the NF-κB:E2F-1 mathematical model. Simulation protocol of a live cell imaging experiment involving transfection and TNFα stimulation.

**Appendix 1—table 1.** A Initial conditions for the NF-κB:E2F mathematical model. Concentrations of NF-κB and E2F-1 added to mimic cell transfection are also included.

| Species | Biological name | Initial Conditions Equilibration stage(μM) | Initial Conditions TNFα stimulation (μM) |
|---|---|---|---|
| NFkB | Cytoplasmic RelA | 0 | 0.004 (+0.1) |
| nNFkB | Nuclear RelA | 0 | 0.015 |
| E2F1 | Cytoplasmic E2F1 | 0 | 0 (+0.1) |
| nE2F1 | Nuclear E2F1 | 0 | 0 |
| tIkBa | IκBα mRNA | 0 | 1e-005 |
| IkBa | Cytoplasmic IκBα | 0 | 0.017 |
| nIkBa | Nuclear IκBα | 0 | 0.004 |
| IKKn | Neutral IKK | 0.1 | 0.1 |
| IKK | Active IKK | 0 | 0 |
| IKKi | Inactive IKKi | 0 | 0 |
| tA20 | A20 mRNA | 0 | 1e-005 |
| A20 | A20 | 0 | 0.001 |
| pIkBa | phospho-IκBα | 0 | 0 |
| pIkBaNFkB | phospho-IκBα RelA complex | 0 | 0 |
| NFkBE2F1 | cyto. RelA E2F-1 complex | 0 | 0 |
| nNFkBE2F1 | nuclear RelA E2F-1 complex | 0 | 0 |
| IkBaNFkB | cyto IκBα RelA complex | 0.1 | 0.091 |
| nIkBaNFkB | nuclear IκBα RelA complex | 0 | 0.001 |
| tE2F4 | E2F-1 target E2F-4 mRNA | 0 | 0 |
| E2F4 | E2F-1 target E2F-4 | 0 | 0 |
| E2F4NFkB | E2F-4 RelA complex | 0 | 0 |
| E2F4IkBaNFkB | E2F-4 IκBα RelA complex | 0 | 0 |

**Appendix 1—table 2.** NF-κB:E2F-1 model equations Symbol 'n' denotes nuclear variables,'t' denotes mRNA transcripts, 'p' denotes phosphorylated form of IκBα. Symbols denoting cytoplasmic localisation were omitted.

$$\begin{aligned}
\frac{d}{dt}NF\kappa B(t) = &-ka1a \times I\kappa B\alpha(t) \times NF\kappa B(t) + kd1a \times (I\kappa B\alpha : NF\kappa B)(t) - ki1 \times NF\kappa B(t) \\
&+ke1 \times nNF\kappa B(t) + kt2a \times (pI\kappa B\alpha : NF\kappa B)(t) + c5a \times (I\kappa B\alpha)(t) \\
&+kd2e \times (NF\kappa B : E2F1)(t) - ka2e \times E2F1(t) \times NF\kappa B(t) + c8ne \times (NF\kappa B : E2F1)(t) \\
&-ka3e \times NF\kappa B(t) \times E2F4(t) + kd3e \times (E2F4 : NF\kappa B)(t) + c4x \times (E2F4 : NF\kappa B)(t)
\end{aligned} \tag{1}$$

$$\begin{aligned}
\frac{d}{dt}nNF\kappa B(t) = &+ka1a \times nI\kappa B\alpha(t) \times nNF\kappa B(t) + kd1a \times (nI\kappa B\alpha : nNF\kappa B)(t) \\
&+ki1 \times kv \times NF\kappa B(t) - ke1 \times kv \times nNF\kappa B(t) \\
&+kd2e \times (nNF\kappa B : nE2F1)(t) - ka2e \times nE2F1(t) \times nNF\kappa B(t) + c9ne \times (nNF\kappa B : nE2F1)(t)
\end{aligned} \tag{2}$$

$$\begin{aligned}
\frac{d}{dt}E2F1(t) = &-kie \times E2F1(t) - kee \times kv \times nE2F1(t) - c6e \times E2F1(t) \\
&+kd2e \times (NF\kappa B : E2F1)(t) - ka2e \times nE2F1(t) \times NF\kappa B(t) \\
&+kdis \times (NF\kappa B : E2F1)(t) \times I\kappa B\alpha(t)
\end{aligned} \tag{3}$$

*Appendix 1—table 2 continued on next page*

$$\frac{d}{dt}nE2F1(t) = +kie \times kv \times E2F1(t) - kee \times kv \times nE2F1(t) - c7e \times nE2F1(t) \\ +kd2e \times (nNF\kappa B : nE2F1)(t) - ka2e \times nE2F1(t) \times nNF\kappa B(t) \\ +kdis \times (nNF\kappa B : nE2F1)(t) \times nI\kappa B\alpha(t) \tag{4}$$

$$\frac{d}{dt}tI\kappa B\alpha(t) = +c1a \times \frac{nNF\kappa B^h(t)}{nNF\kappa B^h(t)+k^h} - c3a \times tI\kappa B\alpha(t) \tag{5}$$

$$\frac{d}{dt}I\kappa B\alpha(t) = kd1a \times (I\kappa B\alpha : NF\kappa B)(t) - ka1a \times I\kappa B\alpha(t) \times NF\kappa B(t) + c2a \times tI\kappa B\alpha(t) \\ -c4a \times I\kappa B\alpha(t) - ki3a \times I\kappa B\alpha(t) + ke3a \times nI\kappa B\alpha(t) - kc1a \times IKK(t) \times I\kappa B\alpha(t) \\ -kdis \times (NF\kappa B : E2F1)(t) \times I\kappa B\alpha(t) - ka3e \times (NF\kappa B : E2F4)(t) \times I\kappa B\alpha(t) \\ +kd3e \times (E2F4 : I\kappa B\alpha : NF\kappa B)(t) \tag{6}$$

$$\frac{d}{dt}nI\kappa B\alpha(t) = kd1a \times (nI\kappa B\alpha : nNF\kappa B)(t) - ka1a \times NI\kappa B\alpha(t) \times nNF\kappa B(t) \\ -c4a \times nI\kappa B\alpha(t) + ki3a \times kv \times I\kappa B\alpha(t) - ke3a \times kv \times nI\kappa B\alpha(t) \\ -kdis \times (nNF\kappa B : nE2F1)(t) \times nI\kappa B\alpha(t) \tag{7}$$

$$\frac{d}{dt}IKKn(t) = kp \times \left(\frac{kbA20}{kbA20+TRA20 \times A20(t)}\right) \times IKKi(t) - TR \times ka \times IKKn(t) \tag{8}$$

$$\frac{d}{dt}IKK(t) = TR \times ka \times IKKn(t) - ki \times IKK(t) \tag{9}$$

$$\frac{d}{dt}IKKi(t) = ki \times IKK(t) - kp \times \frac{kbA20}{kbA20+TRA20 \times A20(t)} \times IKKi(t) \tag{10}$$

$$\frac{d}{dt}tA20(t) = +c1 \times \frac{nNF\kappa B^h(t)}{nNF\kappa B^h(t)+k^h} - c3 \times tA20(t) \tag{11}$$

$$\frac{d}{dt}A20(t) = c2 \times tA20(t) - c4 \times A20(t) \tag{12}$$

$$\frac{d}{dt}pI\kappa B\alpha(t) = kc1a \times IKK(t) \times I\kappa B\alpha(t) - kt1\alpha \times pI\kappa B\alpha(t) \tag{13}$$

$$\frac{d}{dt}(pI\kappa B\alpha : NF\kappa B)(t) = \kappa c2\alpha \times IKK(t) \times (I\kappa B\alpha : NF\kappa B)(t) - \kappa t2\alpha \times (pI\kappa B\alpha : NF\kappa B)(t) \tag{14}$$

*Appendix 1—table 2 continued on next page*

$$\frac{d}{dt}(NF\kappa B : E2F1)(t) = ka2e \times E2F1(t) \times NF\kappa B(t) - kd2e \times (NF\kappa B : E2F1)(t)$$
$$-kine \times (NF\kappa B : E2F1)(t) + kene \times (nNF\kappa B : nE2F1)(t) - c8ne \times (NF\kappa B : E2F1)(t)$$
$$-kdis \times (NF\kappa B : E2F1)(t) \times I\kappa B\alpha(t)$$

(15)

$$\frac{d}{dt}(nNF\kappa B : nE2F1)(t) = ka2e \times nE2F1(t) - nNF\kappa B(t) - kd2e \times (nNF\kappa B : nE2F1)(t)$$
$$+kine \times kv \times (NF\kappa B : E2F1)(t) - kene \times kv \times (nNF\kappa B : nE2F1)(t)$$
$$-c9ne \times (nNF\kappa B : nE2F1)(t) - kdis \times (nNF\kappa B : nE2F1)(t) \times nI\kappa B\alpha(t)$$

(16)

$$\frac{d}{dt}(I\kappa B\alpha : NF\kappa B)(t) = ka1a \times I\kappa B\alpha(t) \times NF\kappa B(t) - kd1a \times (I\kappa B\alpha : NF\kappa B)(t)$$
$$-c5a \times (I\kappa B\alpha : NF\kappa B)(t) + ke2a \times (nI\kappa B\alpha : nNF\kappa B)(t)$$
$$-kc2a \times IKK(t) \times (I\kappa B\alpha : NF\kappa B)(t) + (kdis) \times (NF\kappa B : E2F1)(t) \times I\kappa B\alpha(t)$$
$$-ka3e \times (I\kappa B\alpha : NF\kappa B)(t) \times E2F4(t) + kd3e \times (E2F4 : I\kappa B\alpha : NF\kappa B)(t)$$
$$+c4x \times (E2F4 : I\kappa B\alpha : NF\kappa B)(t)$$

(17)

$$\frac{d}{dt}(nI\kappa B\alpha : nNF\kappa B)(t) = ka1a \times nI\kappa B\alpha(t) \times nNF\kappa B(t) - kd1a \times (nI\kappa B\alpha : nNF\kappa B)(t)$$
$$-ke2a \times kv \times (nI\kappa B\alpha : nNF\kappa B)(t) + kdis \times (nNF\kappa B : nE2F1)(t) \times nI\kappa B\alpha(t)$$

(18)

$$\frac{d}{dt}tE2F4(t) = +clx \times \frac{nE2F1^h(t)}{nE2F1^h(t) + k^h} - c3x \times tE2F4(t)$$

(19)

$$\frac{d}{dt}E2F4(t) = c2x \times tE2F4(t) - c4x \times E2F4(t) - ka3e \times NF\kappa B(t) \times E2F4(t)$$
$$+kd3e \times (E2F4 : NF\kappa B)(t) - ka3e \times E2F4(t) \times (I\kappa B\alpha : NF\kappa B)(t)$$
$$+kd3e \times (E2F4 : I\kappa B\alpha : NFkB)(t)$$

(20)

$$\frac{d}{dt}(E2F4 : NF\kappa B)(t) = ka3e \times NF\kappa B(t) \times E2F4(t) - kd3e \times (E2F4 : NF\kappa B)(t)$$
$$-c4x \times (E2F4 : NF\kappa B)(t) + c5a \times (E2F4 : I\kappa B\alpha : NF\kappa B)(t)$$
$$-ka3e \times (NF\kappa B : E2F4)(t) \times I\kappa B\alpha(t) + kd3e \times (E2F4I\kappa B\alpha NF\kappa B)(t)$$

(21)

$$\frac{d}{dt}(E2F4 : I\kappa B\alpha : NF\kappa B)(t) = +ka3e \times (I\kappa B\alpha : NF\kappa B)(t) \times E2F4(t)$$
$$-kd3e \times (E2F4 : I\kappa B\alpha : NF\kappa B)(t) - c5a \times (E2F4 : I\kappa B\alpha : NF\kappa B)(t)+$$
$$ka3e \times (NF\kappa B : E2F4)(t) \times I\kappa B\alpha(t) - kd3e \times (E2F4 : I\kappa B\alpha : NF\kappa B)(t)$$
$$-c4x \times (E2F4 : I\kappa B\alpha : NF\kappa B)(t)$$

(22)

**Appendix 1—table 3.** Model reactions and associated parameters.

| Reaction | Symbol | Value | References |
|---|---|---|---|
| *Spatial parameters* | | | |
| Total cell volume | tv | 2700 μm³ | Measured |
| C:N ratio | kv | 3.3 | Measured |

*Appendix 1—table 3 continued on next page*

*Appendix 1—table 3 continued*

| Reaction | Symbol | Value | References |
|---|---|---|---|
| Conversion to nuclear volume | nv | $\times(kv+1)$ | - |
| Conversion to cytoplasmic volume | cv | $\times(1/kv+1)$ | - |
| *Initial concentration* | | | |
| Total NF-κB | NF | 0.08 μM | Initialized as cytoplasmic IκBα·NF-κB |
| Total IKK | - | 0.08 μM | Initialized as IKKn |
| *Complex formation & dissociation* | | | |
| IκBα + NF-κB → IκBα·NF-κB<br>nIκBα + nNF-κB → nIκBα·NF-κB | ka1a | $0.5\ \mu M^{-1}s^{-1}$ | (*Hoffmann et al., 2002*) |
| IκBα·NF-κB → IκBα + NF-κB<br>nIκBα·nNF-κB → nIκBα + nNF-κB | kd1a | $0.0005\ s^{-1}$ | (*Hoffmann et al., 2002*) |
| NF-κB + E2F (1 or 4) → NF-κB·E2F<br>nNF-κB + nE2F → nNF-κB·nE2F | ka2e | $0.5\ \mu M^{-1}s^{-1}$ | fitted, same as IκBα + NF-κB |
| NF-κB·E2F → NF-κB + E2F<br>nNF-κB·nE2F → nNF-κB + nE2F | kd2e | $0.0005\ s^{-1}$ | fitted, same as IκBα + NF-κB |
| NF-κB·E2F1 + IκBα → IκBα·NF-κB + E2F1<br>nNF-κB·nE2F1 + nIκBα → nIκBα·NF-κB + nE2F1 | kdis | $0.001\ s^{-1}$ | fitted |
| *Transport* | | | |
| NF-κB → nNF-κB | ki1 | $0.0026\ s^{-1}$ | Measured fitting range:<br>*Average $0.0026 \pm 0.0018 s^{-1}$* |
| nNF-κB → NF-κB | ke1 | $0.000052\ s^{-1}$ | ki1/50 (*Carlotti et al., 2000*) |
| E2F1 → nE2F1 | kie | $0.0026\ s^{-1}$ | fitted, same as NF-κB |
| nE2F1 → E2F1 | kee | $0.000052\ s^{-1}$ | fitted, same as NF-κB |
| IκBα → nIκBα | ki3a | $0.00067\ s^{-1}$ | Measured fitting range:<br>Average $0.00043 \pm 0.00024\ s^{-1}$ |
| nIκBα → IκBα | ke3a | $0.000335\ s^{-1}$ | ki3a/2 (*Carlotti et al., 2000*) |
| nIκBα·nNF-κB → IκBα·NF-κB | ke2a | $0.01\ s^{-1}$ | Fitted |
| NF-κB·E2F1 → nNF-κB·nE2F1 | kine | $0.0026\ s^{-1}$ | fitted, same as NF-κB |
| nNF-κB·nE2F1 → NF-κB·E2F1 | kene | $0.000052\ s^{-1}$ | fitted, same as NF-κB |
| *Protein synthesis & degradation* | | | |
| nNF-κB → nNF-κB + tIκBα<br>Order of hill function, h=2<br>Half-max constant, k=0.065[h](fitted) | c1a | $1.4\times10^{-7}$ $\mu M^{-1}s^{-1}$ | Fitted (constrained):<br>$1.07\times10^{-7} - 8.2\times10^{-7}\mu M^{-1}s^{-1}$<br>(*Femino et al., 1998*);<br>(*Cheong et al., 2006*) |
| tIκBα → tIκBα + IκBα | c2a | $0.5\ s^{-1}$ | (*Lipniacki et al., 2004*) |
| NF-κB·IκBα → NF-κB | c5a | $0.000022\ s^{-1}$ | (*Pando and Verma, 2000*;<br>*Mathes et al., 2008*) |
| nNF-κB·nIκBα → nNF-κB | - | $0\ s^{-1}$ | Assumed (*O'Dea et al., 2007*;<br>*Mathes et al., 2008*) |
| nNF-κB → nNF-κB + tA20<br>Order of hill function, h=2<br>Half-max constant, k=0.065[h] | c1 | $1.4\times10^{-7}$ $\mu M^{-1}s^{-1}$ | Assumed to be the same as IκBα |
| nE2F1 → nE2F1 + tE2F4<br>Order of hill function, h=2<br>Half-max constant, k=0.065[h] | c1x | $9.8\times10^{-7}$ $\mu M^{-1}s^{-1}$ | Fitted |
| tA20 → tA20 + A20 | c2 | $0.5\ s^{-1}$ | - |

*Appendix 1—table 3 continued on next page*

*Appendix 1—table 3 continued*

| Reaction | Symbol | Value | References |
|---|---|---|---|
| tE2F-4→ tE2F-4 + E2F4 | c2x | 0.5 s$^{-1}$ | - |
| tIκBα→ Sink | c3a | 0.0003 s$^{-1}$ | Fitted (constrained): 0.00077-0.00029 s$^{-1}$ (**Blattner et al., 2000**) |
| tA20→ Sink | c3 | 0.00048 s$^{-1}$ | Fitted, constrained >tIκBα turnover (**Ashall et al., 2009**) |
| tE2F4→ Sink | c3x | 0.00048 s$^{-1}$ | Fitted |
| IκBα→ Sink | c4a | 0.0005 s$^{-1}$ | Fitted (constrained): 0.000105 – 0.002 s$^{-1}$ (**Pando and Verma, 2000**; **O'Dea et al., 2007**; **Mathes et al., 2008**) |
| A20 → Sink | c4 | 0.0045 s$^{-1}$ | Fitted |
| E2F4 → Sink | c4x | 0.00016 s$^{-1}$ | Fitted |
| E2F1 → Sink | c6e | 0.00016 s$^{-1}$ | Fitted |
| nE2F1 → Sink | c7e | 0.00016 s$^{-1}$ | Fitted |
| NF-κB·E2F1 → Sink | c8ne | 0.00016 s$^{-1}$ | Fitted |
| nNF-κB·nE2F1 → Sink | c9ne | 0.00016 s$^{-1}$ | Fitted |
| *TNFα stimulation* | | | |
| TNFα | TR | 1/0 | on/off (**Lipniacki et al., 2004**) |
| *IKK parameters* | | | |
| IKKn → IKKa | ka | 0.004 s$^{-1}$ | Fitted, as above |
| IKKa → IKKi | ki | 0.003 s$^{-1}$ | Fitted, as above |
| IKKi → IKKn | kp | 0.0006 s$^{-1}$ | Fitted |
| A20 inhibition rate constant | kbA20 | 0.0018 | Fitted, scales *kp* dependent on receptor state *kbA20×TR* |
| IKKa + IκBα → pIκBα | kc1a | 0.074 s$^{-1}$ | Assumed (0.037×2) (**Heilker et al., 1999**) |
| IKKa + IκBα·NF-κB → pIκBα·NF-κB | kc2a | 0.37 s$^{-1}$ | Assumed (0.037×5×2) (**Heilker et al., 1999**; **Zandi and Karin, 1999**) |
| pIκBα → Sink | kt1a | 0.1 s$^{-1}$ | Fitted |
| pIκBα·NF-κB → NF-κB | kt2a | 0.1 s$^{-1}$ | Fitted |

**Appendix 1—table 4.** Simulation protocols used throughout the manuscript. TNFα stimulation is invoked via TR=0/1. E2F refers to levels of 'transfection' (in μM). E2F4 off/on refers to whether its transcription is switched on or off.

| Figure | Model conditions |
|---|---|
| 3E | TR=0, E2F1= (0.05, 0.1, 0.15), E2F4 off |
| 4A | TR=1, E2F1 = 0.1, E2F4 off |
| 4C | TR=1, E2F1 = 0.1, E2F4 on |
| 4I (G1, G2) | TR=1, E2F1= 0, E2F4 on (but unaffected) |
| 4I (G1/S) | TR=1, E2F1 = 0.2, E2F4 on |
| 4I (S) | TR=1, NFkBIkBa = 0.1, E2F1 = 0, E2F4= 0.1 |

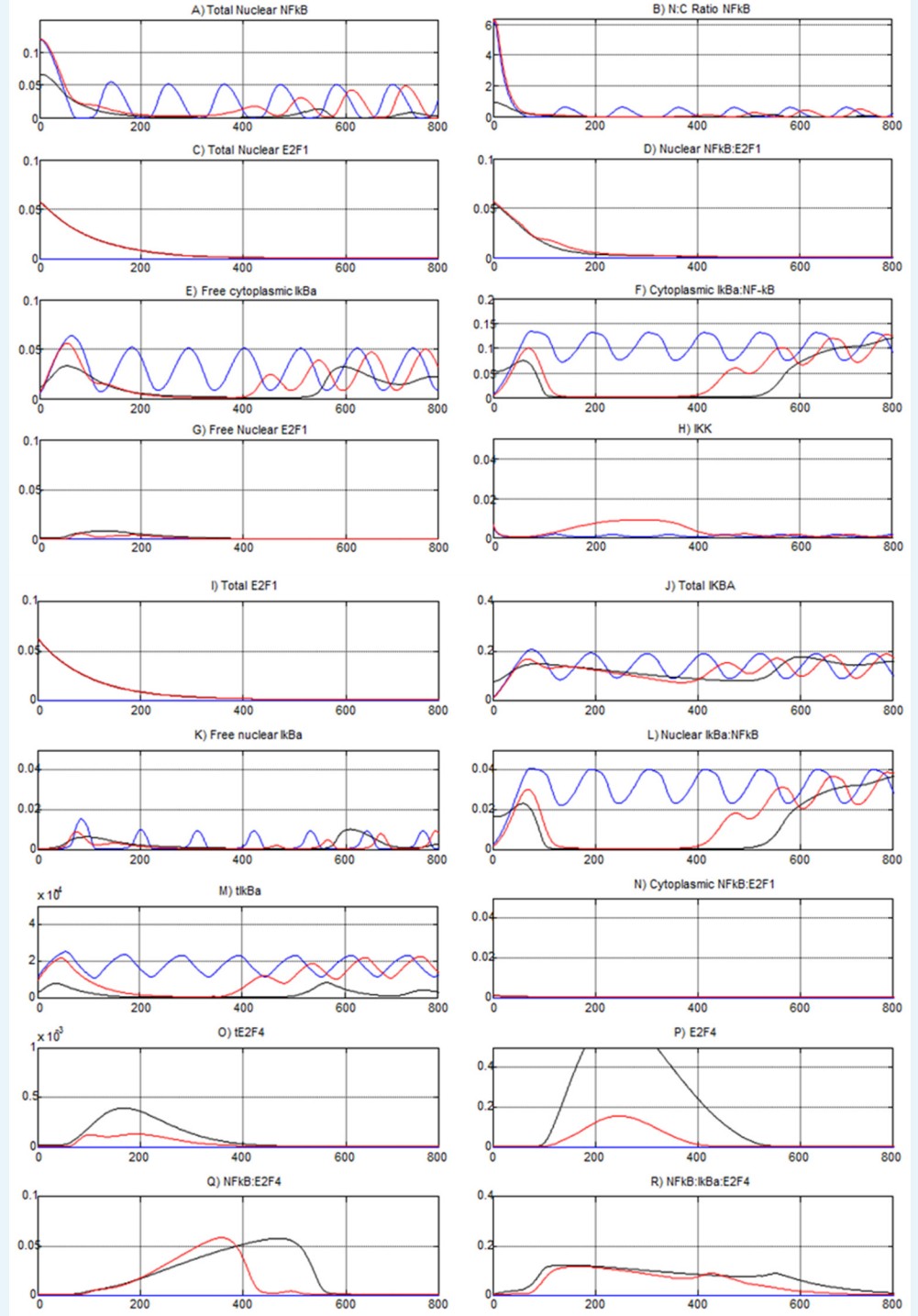

**Appendix 1—figure 6.** Simulations from the NF-κB:E2F model. Blue lines represent *E2F1*= 0, TR= 1. Black lines represent *E2F1= NFkB*= 0.1 and TR=0. Red lines represent *E2F1= NFkB*= 0.1 and TR=1.

## Section C: Generation of recombinant Bacterial Artificial Chromosome (BAC)

Bacterial artificial chromosomes (BACs) containing the human E2F-1 gene with 98.7 kb 5′ flanking DNA, 45.5 kb 3′ flanking DNA (RP11-246G11) or the human RelA gene with 82 kb 5′ and 15 kb 3′ flanking DNA (CTD 2116H8) were identified using genome browser (http://genome.ucsc.edu) and obtained from Invitrogen/Life technologies. The targeting strategy to create BACs expressing fusion protein products were based on the seamless recombineering technology developed by *Warming et al. (2005)*, with minor modifications. Chimeric primers to amplify the GalK gene tagged with homology arms corresponding to the 50–80 bp immediately up and downstream of the stop codon of the gene were used to generate the H-GalK-H recombination cassette for primary targeting. The length of GalK specific portion of the primers was extended to increase Tm and thus the efficiency of the PCR using a two-step PCR method (Phusion high fidelity enzyme, Finnzymes; primer sequences below). The second targeting cassette was generated using chimeric primers with the same homology arms but amplifying the desired fluorescent fusion protein, H-Venus-H for E2F1 or DsRedxp for RelA. Recombination and selection was carried out according to routine protocols (*Warming et al., 2005*) available at http://recombineering.ncifcrf.gov. Clone screening was performed by pulsed field gel electrophoresis, Southern blots and sequencing to confirm in-frame C-terminal insertion of the reporter gene.

Primers used (italics denote homology arm sequence):

E2F1-GalKF

*TCAGAGACCTCTTCGACTGTGACTTTGGGGACCTCACCCCCCTGGATTTC*CCTGTTGACAATTAATCATCGGCATAGTATATCG

E2F1-Galk R

*TGCAGAGACAAGGTGAGCATCTCTGGAAACCCTGGTCCCTCCAAGCCCTG*TCAGCACTGTCCTGCTCCTTGTGA

E2F1-Venus F

*CCACTTCGGCCTCGAGGAGGGCGAGGGCATCAGAGACCTCTTCGACTGTGACTTTGGGGACCTCACCCCCCTGGATTT*CATGGTGAGCAAGGGCGAGGAG

E2F1-Venus R

*CGGCCAGGGACAGGGGGCTCCAGGGCTGCAGAGACAAGGTGAGCATCTCTGGAAACCCTGGTCCCTCCAAGCCCTG*CTACTTGTACAGCTCGTCCATGCC

RelA-GalKF

*ATGAAGACTTCTCCTCCATTGCGGACATGGACTTCTCAGCCCTGCTGAGTCAGATCAGCTCC*CCTGTTGACAATTAATCATCGGCATAGTATATCG

RelA-Galk R

*CAGAATCCGTAAGTGCTTTTGGAGGGCTTCAATCCCCTGCAACCCAGTGCTCTGGGGAGGGCAGGCGTCACCCCC*TCAGCACTGTCCTGCTCCTTGTGA

RelA-DsRedxp F

*ATGAAGACTTCTCCTCCATTGCGGACATGGACTTCTCAGCCCTGCTGAGTCAGATCAGCTCC*ATGGCCTCCTCCGAGGACGTC

RelA-DsRedxp

R*CAGAATCCGTAAGTGCTTTTGGAGGGCTTCAATCCCCTGCAACCCAGTGCTCTGGGGAGGGCAGGCGTCACCCCC*TACAGGAACAGGTGGTGGCG

The BAC were initially transiently transfected into HeLa cells using ExGen500 transfection reagent (Fermentas, UK) to confirm fluorescent protein function.

Ankers *et al.* eLife 2016;5:e10473. DOI: 10.7554/eLife.10473

## BAC stable cell line generation

To generate stable cell lines it was necessary to retrofit the BAC with an appropriate mammalian selection marker. Retrofitting constructs that could universally be applied to any BAC were developed. As the same parent BAC vector construct, pBAC108L, was used to derive the most common BAC vectors, pe3.6 (from the Roslin Park institute library) and the pBeloBAC vector (from the California Institute of Technology) approximately 6kb of the vectors had perfect sequence homology. Within this region the chloramphenicol resistance gene was identified as a suitable target for replacement with a new selection marker as this would not disrupt important bacterial sequences. Restriction site-tagged homology arms 300-400 bp in length were amplified from the BAC sequence using the primers (underlined indicates enzyme site):

5'H *Kpn*I F tgtcaaGGTACCGGCAGCCACATCCAG,

5'H *Eco*RI R ggtgccGAATTCTCAACGTCTCATTTTCGC,

3'H *Bam*HI F aatgggGGATCCTGGACAACTTCTTCGCC,

3'H *Sac*II R aatgggCCGCGGGCCGTCGACCAATTCTC

and cloned using the appropriate enzymes into the multiple cloning sites of pL451 (*Liu et al., 2003*). This resulted in a recombination cassette containing H-pGK-pEM7-Kan/Neo-H. Retrofitting was performed in the same SW102 strain hosting the BAC by heat induction of the bacteria for recombination, transforming with the cassette and plating on LB containing Kanamycin (25 µg/ml). Clones were screened by PFGE and >90% recombination efficiency was observed.

## Stable BAC transfection

BAC DNA was prepared by maxiprep (BAC100 Nucleobond kit, Macherey-Nagel, Germany) and 1 µg or 3 µg used to transfect $10^6$ cells in a 10 cm dish using ExGen500 transfection reagent. Media was changed 3 days post transfection and supplemented with 500 µg/ml G418. Media + antibiotic were refreshed every 3–4 days. Colonies formed 2–3 weeks after culturing in selection containing media were ring cloned into individual wells of a 48 well plate and sequentially scaled up to large culture vessels as necessary.

A HeLa cell line stably expressing the E2F-1-venus bacterial artificial chromosome (BAC), was transiently transfected with a FUCCI marker for G1-phase. Shows two consecutive cycles of HeLa cell division for a representative cell (*Figure 8—figure supplement 1*). Parent and daughter cells showed cycles of E2F-1 expression, with a peak timing consistent with late G1-phase.

## Section D: BAC characterisation

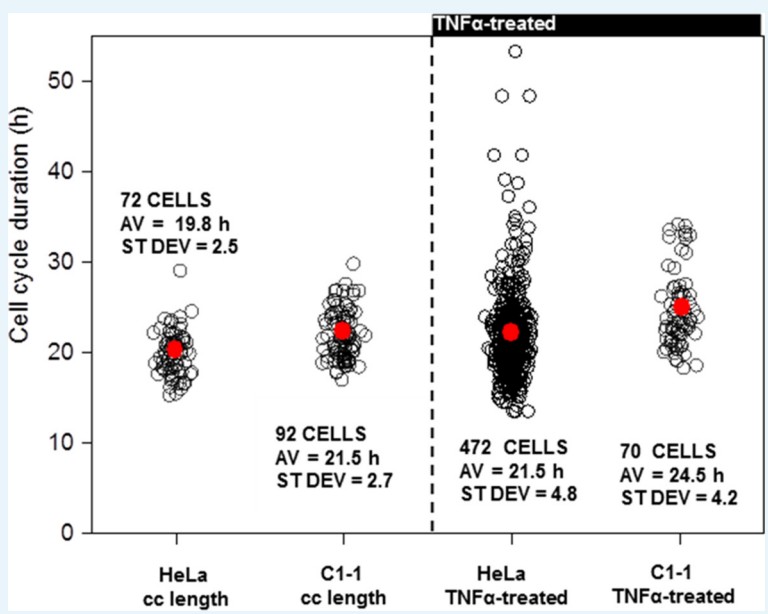

**Appendix 1—figure 7.** Cell Cycle length of Clonal HeLa BAC population. (**A**) Analysis of cell cycle duration in populations of dual BAC stable cell line (C1-1) with wild type HeLa cells. (**B**) Analysis of the effects of TNFα treatment in C1-1 and WT HeLa cells.

## Section E: Controls for FCCS

Physical interaction between RelA and IκBα was confirmed using FCCS in SK-N-AS cells co-expressing RelA-DsRedxp and IκBα-EGFP. FCS exploits fluorescence-intensity fluctuations caused by low numbers of diffusing labelled particles in a diffraction limited confocal volume of light to analyse their concentration and mobility (*Spiller et al. 2010*). Fluctuations are recorded as function of time and then statistically analysed by autocorrelation analysis. In its dual-colour variant, FCCS, two spectrally distinct fluorophores (such as red and green) are used and the cross-correlation amplitude in conjunction with the auto-correlation amplitudes provides information on molecular binding as well as dynamic co-localisation. In contrast to FRET, FCCS does not depend on the very close proximity of the interacting fluorescent labels.

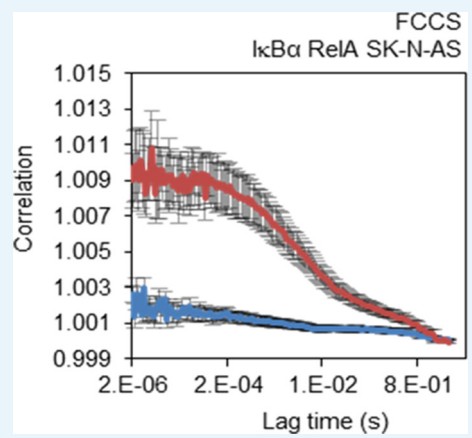

**Appendix 1—figure 8.** FCCS control. FCCS assays between transiently transfected RelA-

dsRedxp and IκBα-EGFP (red line) and Empty-DsRedxp and Empty-EGFP (blue line) in single live SK-N-AS cells (+/- s.e.m based on 10 measurements in each of 10+ cells per condition).

Auto-correlation analysis was conducted to show comparable noise between measurements and between controls for RelA/E2F-1 FCCS data

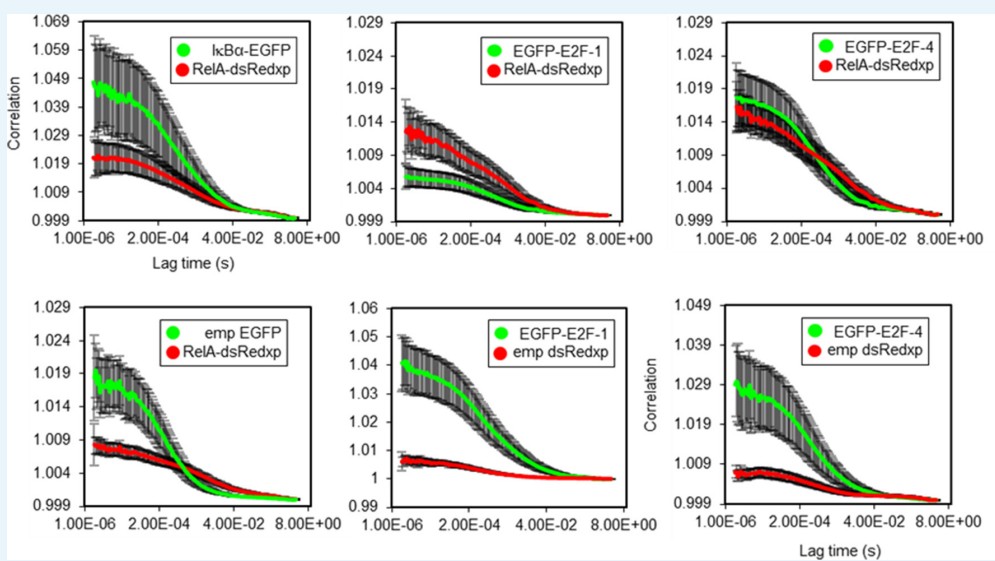

**Appendix 1—figure 9.** FCCS autocorrelation analysis. Autocorrelation lines for RelA/IkBα, RelA/E2F-1 and RelA/E2F-4 FCCS studies in single live SK-N-AS cells (+/- s.e.m based on 10 measurements in each of 10+ cells per condition).

