## [Decision Letter]

Thank you for submitting your work entitled "Dynamic NF-κB and E2F interactions control the priority and timing of inflammatory signalling and cell proliferation" for consideration by *eLife*. Your article has been reviewed by three peer reviewers, and the evaluation has been overseen by a Reviewing Editor and Aviv Regev as the Senior Editor.

The reviewers have discussed the reviews with one another and the Reviewing editor has drafted this decision to help you prepare a revised submission.

Summary:

Ankers et al. examine the impact of the cell cycle on cellular variability during NF-κB response to TNFα. Using a combination of fluorescent microscopy, biochemical assays, and a computation model, they show cell cycle dependent variability in NF-κB response, which is a result of physical interactions between NF-κB with E2F-1 and E2F-4 proteins.

Specifically, using transient overexpression of fluorescent fusion proteins combined with cell cycle synchronization, the authors show that the NF-κB response to TNF is attenuated in S-phase, while pathway activation before G1/S transition prolongs the cell cycle. The refractoriness of cells in S-phase led the authors to test an E2F-1 target gene E2F-4 for a role in mediating the S phase-specific inhibitory interaction of RelA with E2F-4. The authors found that interplay between cell cycle and NF-κB response is based on physical interactions of RelA with the transcription factors E2F-1 and -4, as shown by CO-IP, FRET and FCCS. While E2F-1 leads to prolonged nuclear accumulation of RelA by competing with IκBα binding and preventing expression of the feedback regulator, E2F-4 binding leads to cytoplasmic localization of RelA. Finally, the authors validate their results using fusion proteins expressed at physiological levels through a BAC expression system.

The reported findings are highly significant for understanding the interplay between cellular signalling and cell cycle control and are potentially of interest for a broad audience. However, the reviewers have outlined concerns that needed to be addressed to strengthen the stated conclusions of the manuscript.

Essential revisions:

1) The authors' use of transiently transfected RelA, in Figure 1–Figure 3 weakens the data by two factors: a) RelA abundance is bound to be very variable from cell to cell and b) variable RelA abundance could affect cell cycle timing yet the cell cycle mapping using time since mitosis is based on calibration done in other cells. The data from the BAC stable cell line introduced later in the paper is much stronger. We suggest that the authors either acquire data for a greater cell number with this cell line and replace data in Figure 1–Figure 3 with the new data, or alternatively, integrate data from the BAC cell lines with that of the transiently transfected cell line in the early figures of the paper so that the first segment of the is strengthened.

2) Overall, in the data presented in this manuscript, the oscillations in nuclear RelA fusion protein amount seem very damped with little or no second peaks. Importantly, this is observed not only for HeLa (which showed more damped oscillations in their original 2004 Science paper) but also for SK-N cells (which showed more robust oscillations in the 2004 paper). Although the study focuses on a shorter time window encompassing mostly the first peak, the apparent lack of robust oscillations will be puzzling to many readers familiar with the authors' previous work and warrants some explanation.

3) Some of the experiments require more upfront controls, validations or stronger explanations:

A) The authors should show by immunoblot that synchronization of HeLa cells with a double thymidine block of DNA synthesis does not significantly change the abundance of tagged RelA.

B) How does overexpression of E2F-1 and E2F-4 impact cell cycle? Does inducing NF-κB with TNF during E2F-1 overexpression have a negative impact on cells that are in S-phase and that normally would not respond during this time?

C) The CO-IPs in 5D and 7B would benefit from showing a RelA pull down in each of the other phases as a comparison to the phase in question.

D) In 5B and 7D, the fluorescence changes for YFP and CFP are unexpectedly symmetrical after photobleaching: Because the inter-molecular FRET ratio to CFP is likely only a few percent, how can the increase in CFP be of the same magnitude as the decrease in YFP? Related, why is efficiency of YFP bleaching so different between these two experiments? A better explanation of the experiment and quantification of the results would really help, as would bringing the controls in Section E of the supplements to the main figures (or a linked supplement), allowing the readers to more easily compare results obtained with photobleached non-interacting probes.

4) The consequence of the E2F-1 – RelA interaction is unclear. Overexpression of E2F-1 inhibits RelA target gene expression despite inducing its nuclear accumulation (Figure 4). Is this caused by loss of RelA-binding to DNA or is the recruitment of the transcriptional machinery prevented? As mentioned in the Discussion, previous studies reported that both proteins are binding to the same promoters and co-activate genes (e.g. Lim et al., Mol Cell 2007). How does this fit in with the data in this manuscript and the interpretation? The authors hypothesize that loss of feedback (IκBα) expression causes the stronger NF-κB response at G1-S. While additional experiments may be outside of the scope of the study, overall this interpretation is not intuitive and therefore it is important for the authors to both clarify their interpretation and how it is supported (or not) by the different pieces of evidence.

5) The authors show that binding to E2F-4 keeps RelA in the cytoplasm, however, no data at this time elucidates whether E2F-4 competes for IκBα binding or whether a tertiary complex formed. Furthermore, the immunoblot data shown in Figure 6 is not definitive enough to demonstrate that the effect of E2F-4 is through the stabilization of IκBα, and even so, this interpretation contradicts the final model in Figure 9 which is confusing to readers. While further characterization of these interactions may be outside of the scope of the current study, the model in Figure 9 should be consistent will all the data, and it should be made clear that the role of E2F-4 is speculative. Furthermore, it would be helpful to briefly mention in the main text what terms are included in the refined model for E2F-4: binding to RelA and/or IκBα? This way the readers would appreciate which interactions are necessary to explain the data without delving into the model descriptions in the Supplementary section.

6) The authors show that NF-κB activation at the G1/S boundary induces elongation of the cell cycle. However, it remains unclear what the molecular mechanisms are, and the effects are subtle. The reviewers therefore suggest that authors put overall the greatest focus of the manuscript on the modulation of NF-κB signalling by E2F proteins.

7) As there is existing literature regarding the interaction of NF-κB, E2Fs and the cell cycle – in 2002, Tanaka et al. showed reduced NF-κB activation in S-phase compared to G1 and G2, although upon a different stimulus, and could connect this to the state of E2F-1 protein (Tanaka et al. Mol Cell 2002) – the authors should discuss and cite these results.

---

## [Author Response]

Essential revisions:

1) The authors' use of transiently transfected RelA, in Figure 1–Figure 3 weakens the data by two factors: a) RelA abundance is bound to be very variable from cell to cell and b) variable RelA abundance could affect cell cycle timing yet the cell cycle mapping using time since mitosis is based on calibration done in other cells. The data from the BAC stable cell line introduced later in the paper is much stronger. We suggest that the authors either acquire data for a greater cell number with this cell line and replace data in Figure 1–Figure 3 with the new data, or alternatively, integrate data from the BAC cell lines with that of the transiently transfected cell line in the early figures of the paper so that the first segment of the is strengthened.

The reviewers suggested that Figure 1–Figure 3 be either replaced or integrated with data from the BAC cell line. We feel that the introduction of the BAC cell line early in the paper would adversely affect the flow of the manuscript, since transient transfection is also used in Figure 4, Figure 5, Figure 6 and Figure 7. The reviewers nevertheless suggest that data from the BAC stable cell line is much stronger than the transiently transfected RelA data and that we should include more data from this cell line. We agree that the BAC stable cell line is a significant step forward and strengthens interpretation of the transient transfection data. We have now generated and analysed the data from a substantial number of cells from this cell line. These new data map the level of NF-κB translocation in relation to cell cycle phase for a substantial number of cells. (See new Figure 8). This new data set also presented the opportunity for the first time to map the corresponding E2F1-Venus levels across a population of cells. These data are shown in a new Figure 8—figure supplement 1. The trace of E2F expression at different cell cycle times is also used to help illustrate the new statistical analysis of the complete RelA translocation data set, which is shown in Figure 8. We were able to infer on a cell-by-cell basis the cell cycle phase (G1, S-phase, G2) for each cell at the onset of TNFα stimulation. This was achieved by using the timing of the peak of E2F-1 expression to identify the onset of S-phase. This approach was validated through direct comparison of the E2F-1 trace in single E2F-1-Venus BAC stable cells transfected with FUCCI G1 phase marker construct (now Figure 8—figure supplement 2).

Rather than superseding the transient transfection data, we therefore believe that the data from analysis of the double BAC cell line is entirely complementary with the earlier presentation of the transient transfection data. This order of presentation of the experiments provides a more logical experimental framework in which the BAC cell data can be characterised and interpreted. We hope the reviewers will agree that the new Figure 8 adds a substantial and important new data set that helps to illustrate and support the ordering of the presentation of the data and the optimal development of the evidence for the main conclusions of this paper.

*2) Overall, in the data presented in this manuscript, the oscillations in nuclear RelA fusion protein amount seem very damped with little or no second peaks. Importantly, this is observed not only for HeLa (which showed more damped oscillations in their original 2004 Science paper) but also for SK-N cells (which showed more robust oscillations in the 2004 paper). Although the study focuses on a shorter time window encompassing mostly the first peak, the apparent lack of robust oscillations will be puzzling to many readers familiar with the authors' previous work and warrants some explanation.* The impression that there are more damped or fewer oscillations in nuclear RelA is due (as the reviewers note) to the shorter time window of the experiments. We have now included an additional Figure that illustrates the traces over a longer timecourse (Figure 10). We hope that this will remove the impression that there is a discrepancy in these data.

3) Some of the experiments require more upfront controls, validations or stronger explanations:

A) The authors should show by immunoblot that synchronization of HeLa cells with a double thymidine block of DNA synthesis does not significantly change the abundance of tagged RelA.

We show that a double thymidine block has no effect on RelA-dsRedxp levels in the C1-1 double BAC stable cell line. This blot is now included in the Appendix ( Figure 12).

*B) How does overexpression of E2F-1 and E2F-4 impact cell cycle? Does inducing NF-κB with TNF during E2F-1 overexpression have a negative impact on cells that are in S-phase and that normally would not respond during this time?* Transient transfection of E2F-1 plasmids with a CMV promoter caused apoptosis and cells were rescued by co-transfection with Rel-A. It was unclear if apoptosis was caused by overexpression or out of context expression of E2F-1. In contrast E2F-4 transient transfection did not cause apoptosis and cells were seen to divide as normal. A key attribute of the dual BAC stable cell line was that (as shown in ) the cells express low levels of the E2F-1-Venus fusion protein that are much lower than the endogenous E2F-1 levels. Therefore, the E2F-1 fluorescent protein expression in this context appears to not perturb the normal cell cycle. It must be noted that throughout the virtual synchronisation experiments we analysed data from cells through two successive cell divisions.

*C) The CO-IPs in 5D and 7B would benefit from showing a RelA pull down in each of the other phases as a comparison to the phase in question.* We fully understand this suggestion. However, the CO-IPs were difficult to perform and gave only a low signal. This appeared to be in agreement with the FRET and FCS data which suggested a relatively low affinity interaction between E2F-1 and E2F-4 and RelA. In our experiments we were only able to see any indication of an interaction by CO-IP in cells that had been synchronised in late G1/S (i.e. at the peak of E2F-1 expression). We could not see a significant interaction in asynchronous cells. Under these circumstances showing a negative CO-IP blot at other cell cycle stages seems uninformative. In this context it should be noted that this is not a new piece of data, since other groups had already shown this interaction by pull-down in other cell types (see papers by Lim et al., Garber et al., and Tanaka et al., cited in the manuscript).

*D) In 5B and 7D, the fluorescence changes for YFP and CFP are unexpectedly symmetrical after photobleaching: Because the inter-molecular FRET ratio to CFP is likely only a few percent, how can the increase in CFP be of the same magnitude as the decrease in YFP? Related, why is efficiency of YFP bleaching so different between these two experiments? A better explanation of the experiment and quantification of the results would really help, as would bringing the controls in Section E of the supplements to the main figures (or a linked supplement), allowing the readers to more easily compare results obtained with photobleached non-interacting probes.* The reviewer notes that changes in donor fluorescence appear to be beyond the physical limit of non-radiative energy transfer between CFP and YFP. However, within this manuscript, FRET measurements following acceptor photobleaching were purely qualitative measures of interaction, used to supplement FCCS and CO-IP measurements. Data were collected using λ scans and linear unmixing using reference spectra and no efficiency of interaction was assumed. Measurements were normalised to pre-bleach fluorescence, and the relative change of donor fluorescence was used to qualitatively infer FRET. The manuscript has been amended appropriately and negative controls have been added to Figure 5.

4) The consequence of the E2F-1 – RelA interaction is unclear. Overexpression of E2F-1 inhibits RelA target gene expression despite inducing its nuclear accumulation (Figure 4). Is this caused by loss of RelA-binding to DNA or is the recruitment of the transcriptional machinery prevented? As mentioned in the Discussion, previous studies reported that both proteins are binding to the same promoters and co-activate genes (e.g. Lim et al., Mol Cell 2007). How does this fit in with the data in this manuscript and the interpretation? The authors hypothesize that loss of feedback (IκBα) expression causes the stronger NF-κB response at G1-S. While additional experiments may be outside of the scope of the study, overall this interpretation is not intuitive and therefore it is important for the authors to both clarify their interpretation and how it is supported (or not) by the different pieces of evidence. The reviewers make a good point here that we had tried to discuss in the manuscript. We do not suggest that E2F-1 acts to sequester RelA in the nucleus through strong binding (see response to point 3 above). While this might be suggested by our experiments in Figure 4, it must be recognised that these experiments (although informative) are based on expression from a plasmid vector. Instead we suggest that E2F-1 competes with IκBα binding and the RelA-E2F-1 complex can then interact with promoters to differentially modulate transcription.

As shown in Figure 4, the predominantly cytoplasmic localisation of RelA-dsRedxp when expressed alone was changed to a predominantly nuclear localisation in cells co-expressing RelA-dsRedxp and EGFP-E2F-1. In addition however we also find that the steady-state cytoplasmic localisation of RelA is restored in cells expressing IkBα-AmCyan, EGFP-E2F-1 and RelA-dsRedxp (new panel now added to Figure 4). These data suggest the hypothesis that IkBα and E2F-1 compete for the same binding site on RelA, with IkBα perhaps having the higher affinity. We therefore suggest that at G1/S the high E2F1 levels result in formation of a transcription factor complex between RelA and E2F-1 in the nucleus of TNFα stimulated cells. This might activate some genes and repress others. However, our data in Figure 4 suggest that the presence of E2F-1 results in repression of NF-κB-dependent IκBα expression. As a result, we suggest that the longer and often larger peak of nuclear RelA in G1/S simulated cells may be due to a delay in the new round of IκBα expression and synthesis. The mathematical model fully encapsulates this hypothesis. We have amended the Discussion to try to make these points clearer.

5) The authors show that binding to E2F-4 keeps RelA in the cytoplasm, however, no data at this time elucidates whether E2F-4 competes for IκBα binding or whether a tertiary complex formed. Furthermore, the immunoblot data shown in Figure 6 is not definitive enough to demonstrate that the effect of E2F-4 is through the stabilization of IκBα, and even so, this interpretation contradicts the final model in Figure 9 which is confusing to readers. While further characterization of these interactions may be outside of the scope of the current study, the model in Figure 9 should be consistent will all the data, and it should be made clear that the role of E2F-4 is speculative. Furthermore, it would be helpful to briefly mention in the main text what terms are included in the refined model for E2F-4: binding to RelA and/or IκBα? This way the readers would appreciate which interactions are necessary to explain the data without delving into the model descriptions in the Supplementary section. Our data proposes a clear mechanism for the functional action of E2F-1. However, the mechanism of E2F-4 remains less clear. One issue that must be recognised is that these are just two members of a family of E2F proteins. It has been outside the feasible scope of the present study to characterise them all. One surprising characteristic of the E2F-4 transcription factor is that its predominant localisation (at least in some cell types) is in the cytoplasm. Presumably a subset of the protein is also in the nucleus where it acts as a transcription factor. In the response to the question about E2F-1 above we have added to the paper data that supports the hypothesis that E2F-1 may compete with IκBα for binding to RelA. This was possible due to the predominant nuclear localisation of E2F-1. However due to its cytoplasmic localisation a similar experiment is not possible for E2F-4. In FCS and FRET experiments we found no evidence for an interaction between either E2F-1 and IκBα or E2F-4 and IκBα (data not shown). Therefore, it seems plausible, as suggested in the comment by the reviewers, that E2F-4 may also compete for IκBα binding. However in the absence of an E2F-4 BAC cell line (which was beyond the scope of this study) it is not possible to provide any clear evidence for the mechanism of E2F-4 inhibition. The mathematical model (and the diagram in Figure 9) for simplicity currently has E2F-4 binding to the RelA-IκBα complex. We have endeavoured to include more discussion in the paper that makes this issue clearer, so that readers will understand the extent of our current evidence and claims.

The representation in Figure 9 is entirely in agreement with our data and our model and is currently the simplest representation of the system, given the uncertainties discussed above. We had included a phosphorylation on the RelA molecule in the diagram that was a distraction and this has now been removed. We hope that this may make the diagram clearer to the readers.

*6) The authors show that NF-κB activation at the G1/S boundary induces elongation of the cell cycle. However, it remains unclear what the molecular mechanisms are, and the effects are subtle. The reviewers therefore suggest that authors put overall the greatest focus of the manuscript on the modulation of NF-κB signalling by E2F proteins.* In the manuscript, we reported how TNFα treatment results in the elongation of cell cycle. We further noted that this elongation is dependent upon the cell cycle phase with cells treated in late G1 being more susceptible to elongation and cells treated in S-phase showing no difference from the untreated population average. When comparing the results of late G1 to S-phase treated cells, the differences are quite dramatic and statistically significant. Since we also saw high NF-κB response in late G1 and a reduced response in S-phase, we suggested that there is a 2-way interaction between the NF-κB and cell cycle systems. In agreement with the reviewers, we note that this remains a hypothesis and that we do not give a specific mechanism. We believe these data are interesting and are indicative of the potential direct role of NF-κB in mediating cell cycle duration following cytokine stimulation.

Following the reviewers’ suggestion we have reworded the corresponding Results section in the manuscript to better clarify our observation. In addition, we have expanded the Discussion to explore possible hypotheses regarding this observation and now include mention of other possible downstream cell cycle targets of NF-κB. We have also amended the final sentence of the Abstract so that the main focus is on cell cycle control of NF-κB responses.

7) As there is existing literature regarding the interaction of NF-κB, E2Fs and the cell cycle – in 2002, Tanaka et al. showed reduced NF-κB activation in S-phase compared to G1 and G2, although upon a different stimulus, and could connect this to the state of E2F-1 protein (Tanaka et al. Mol Cell 2002) – the author should discuss and cite these results.

We have now cited this paper and have included discussion of its results.